# INVESTIGATING MULTI-TASK PRETRAINING AND GENERALIZATION IN REINFORCEMENT LEARNING

**Adrien Ali Taïga**
MILA, Université de Montréal
Google Brain
alitaiga@google.com

**Rishabh Agarwal**
MILA, Université de Montréal
Google Brain
rishabhagarwal@google.com

**Jesse Farebrother**
Mila, McGill University
Google Brain
farebroj@mila.quebec

**Aaron Courville** *
MILA, Université de Montréal
aaron.courville@umontreal.ca

**Marc G. Bellemare** *
Google Brain
bellemare@google.com

## ABSTRACT

Deep reinforcement learning (RL) has achieved remarkable successes in complex single-task settings. However, designing RL agents that can learn multiple tasks and leverage prior experience to quickly adapt to a related new task remains challenging. Despite previous attempts to improve on these areas, our understanding of multi-task training and generalization in RL remains limited. To fill this gap, we investigate the generalization capabilities of a popular actor-critic method, IMPALA (Espeholt et al., 2018). Specifically, we build on previous work that has advocated for the use of modes and difficulties of Atari 2600 games as a challenging benchmark for transfer learning in RL (Farebrother et al., 2018; Rusu et al., 2022). We do so by pretraining an agent on multiple *variants* of the same Atari game before fine-tuning on the remaining *never-before-seen* variants. This protocol simplifies the multi-task pretraining phase by limiting negative interference between tasks and allows us to better understand the dynamics of multi-task training and generalization. We find that, given a fixed amount of pretraining data, agents trained with more variations are able to generalize better. Surprisingly, we also observe that this advantage can still be present after fine-tuning for 200M environment frames than when doing zero-shot transfer. This highlights the potential effect of a good learned representation. We also find that, even though small networks have remained popular to solve Atari 2600 games, increasing the capacity of the value and policy network is critical to achieve good performance as we increase the number of pretraining modes and difficulties. Overall, our findings emphasize key points that are essential for efficient multi-task training and generalization in reinforcement learning.

## 1 INTRODUCTION

Deep RL has achieved remarkable results in recent years, from surpassing human-level performance on challenging games (Silver et al., 2017; Berner et al., 2019; Vinyals et al., 2019) to learning complex control policies that can be deployed in the real world (Levine et al., 2016; Bellemare et al., 2020). However, these successes were attained with specialized agents trained to solve a single task and with every new task requiring a new policy trained from scratch. On the other hand, *high-capacity* models trained on large amounts of data have remarkable generalization abilities in other deep learning

---

*CIFAR Fellow

domains such as vision and NLP (Brown et al., 2020; He et al., 2022). Such models can solve multiple tasks simultaneously (Chowdhery et al., 2022), show emergent capabilities (Srivastava et al., 2022), and quickly adapt to unseen but related tasks by leveraging previously acquired knowledge (Brown et al., 2020). While recent work has succeeded in learning such broadly generalizing policies using supervised learning (Reed et al., 2022; Lee et al., 2022), despite many attempts, deep RL agents have not been able to achieve the same kind of broad generalization.

RL policies pretrained on multiple tasks are often unable to leverage information about previous tasks to accelerate learning on related tasks (Kirk et al., 2021). Furthermore, multi-task agents are reported to perform worse on individual tasks than a single-task agent trained on that task (Espeholt et al., 2018). The limited performance of multi-task policies is believed to be due to negative interference between tasks (Schaul et al., 2019). Recent work has tried to address this issue, Hessel et al. (2019) proposes a reward rescaling scheme that normalizes the effect of each task on the learning dynamics. Similarly Guo et al. (2020) introduces a self-supervised learning objective to improve representation learning during multi-task training. Other works have shown that multi-task pretraining can lead to better representations that are useful for downstream tasks when the policy only needs to generalize to a new reward (Borsa et al., 2016; Yang et al., 2020; Sodhani et al., 2021). Nevertheless, in complex multi-task problems, such as the Arcade Learning Environment (ALE; Bellemare et al., 2013), generalization of pretrained policies to unseen tasks remains an unsolved problem.

In this paper, we propose to take a closer look at multi-task RL pretraining and generalization on the ALE, one of the most widely used deep RL benchmark. Though, one might hope that an agent could benefit from multi-task training to learn some high level concepts such as affordances (Khetarpal et al., 2020) or contingency awareness (Bellemare et al., 2012), the lack of common ground between tasks makes joint training on multiple Atari games a difficult problem. To circumvent these issues, we make use of the *modes* and *difficulties* of Atari 2600 games (Figure 1), which we call *variants* (Machado et al., 2018). These *variants* were developed by game designers to make each game progressively more challenging for human players. Previous work has advocated for their use to study generalization in reinforcement learning (Farebrother et al., 2018; Rusu et al., 2022), though these works did not go beyond pretraining on a single task. Notably, Farebrother et al. (2018) argued that the representation learned by a DQN agent (Mnih et al., 2013) after pretraining is brittle and is not able transfer well to *variants* of the same game. They find that during pretraining, the representation tends to overfit to the task it is being trained on, decreasing its generalization capabilities.

Our goal in this work is to revisit these results under a new light, leveraging advances in algorithms, architectures and computation that have happened since. We use a more efficient algorithm, IMPALA (Espeholt et al., 2018) instead of DQN, much larger networks than the decade old 3-layer convolutional neural networks used by DQN (Mnih et al., 2013) and pretrain our agents using multiple *variants* of a game as opposed to just a single one. By limiting the pretraining tasks to different *variants* of the same game, we facilitate multi-task transfer between variants, which in-turn allows us to study under what conditions RL algorithms are able to generalize better when training on multiple tasks. Our contribution are as follow:

- We find that pretrained policies can achieve zero-shot transfer on variants of the same game. If we then fine-tune these polices using interactions in the unseen variant, the fine-tuned policies generalize quite well and learn significantly faster than a randomly initialized policy.

- We observe that a good representation can be learned using pretraining on a relatively small number of modes. Fine-tuning performance from these representations improve as we increase the amount of the pretraining data, as opposed to overfitting on pretraining tasks.

- Finally, we demonstrate that it is possible to train high capacity networks such as residual networks (He et al., 2016), with tens of millions of parameters, using online RL. We find that increased representation capabilities from such networks are essential to reach peak performance in the multi-task regime.

## 2 BACKGROUND

**Reinforcement learning:** We consider a Markov decision process (MDP) (Sutton & Barto, 2018) $M$ represented by a tuple $\langle \mathcal{S}, \mathcal{A}, P, r, \gamma, \rho \rangle$ with $\mathcal{S}$ the state space, $\mathcal{A}$ the finite set of actions, $P : \mathcal{S} \times \mathcal{A} \to [0, 1]^{\mathcal{S}}$ the transition probability distribution, $r : \mathcal{S} \times \mathcal{A} \mapsto \mathbb{R}$ the reward function, $\gamma \in [0, 1]$

| Space Invaders | Asteroids | Breakout | Bank Heist |
|---|---|---|---|

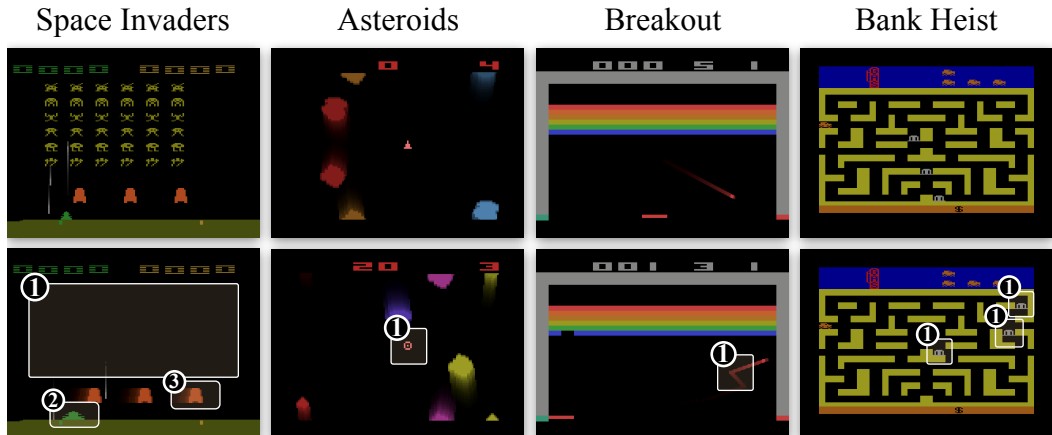

Figure 1: Examples of **game *variants*** of Atari 2600 games. Each column represents two different variants of a game, where we show their visual differences using the numbered rectangles. In SPACE INVADERS shields move from static to mobile, in ASTEROIDS the shape of the spaceship changes, in BREAKOUT the ball becomes steerable by the paddle, in BANK HEIST banks appear at different locations. While there are only subtle changes in the visual observations or agent embodiment, such changes are crucial for the optimal policy, which makes generalization on these *variants* difficult.

the discount factor and $\rho$ the initial state distribution. At each timestep $t$ the agent observe the current state $s_t$, then its chooses an action sampled according to its policy $\pi : \mathcal{S} \times \mathcal{A} \to [0,1]$ before it can receive a reward $r_t$ and observe the subsequent state $s_{t+1}$. Given a policy $\pi$, we define the value function at time $t$ to be the discounted sum of expected rewards to be collected from the current state when following policy $\pi$: $V(s_t) := \mathbb{E}\big[ \sum_{i=0}^{+\infty} \gamma^i r_{t+i} | s_t, a_t \sim \pi(s_t) \big]$. The agent's goal is to find a policy that maximizes its value in each state.

**Transfer learning / Generalization:** In transfer learning, we are interested in designing learning methods that are able to leverage previous interactions with a task / environment to solve new tasks more efficiently. There are many possible ways to quantify transfer in reinforcement learning. The first one we study is zero-shot where performance is evaluated on the target tasks without any adaptation (*e.g.,* Cobbe et al., 2020a; Agarwal et al., 2021a; Zhang et al., 2021). We also evaluate performance after the agent is allowed to interact with the target environment for a fixed number of online environment interactions. We refer the reader to Zhu et al. (2020); Kirk et al. (2021) for a in-depth overview on transfer and generalization in deep reinforcement learning.

**Arcade Learning Environment:** The Arcade Learning Environment (Bellemare et al., 2013) provides an interface to Atari 2600 games and was first proposed as a platform to evaluate the abilities of reinforcement learning agents across a wide variety of environments. It has since become a cornerstone of reinforcement learning research. Practitioners have focused mostly on training agents individually on each game to achieve the highest possible reward and current techniques far exceed human performance (Mnih et al., 2015; Van Hasselt et al., 2016; Bellemare et al., 2017; Kapturowski et al., 2018; Badia et al., 2020). Recent work has also looked at the ALE in the context of generalization, transfer learning and continual learning (Farebrother et al., 2018; Rusu et al., 2022). The major benefit of the ALE compared to other generalization benchmarks (Nichol et al., 2018; Yu et al., 2020) is that it proposes a diverse set of games that is free of researcher bias, as these games were designed to be challenging for human players, rather than for improving deep RL agents. The latest version of the ALE (Machado et al., 2018) introduces different modes and difficulties for Atari 2600 games. These flavours were part of the original titles and were designed to offer players a more entertaining experience by offering variations of the default game. Each variant offers a different alteration of the base game across some discrete modifications while most of the game remains the same (see Figure 1). Because the underlying dynamics of the environment do not change much we can expect that playing some modes can lead to faster learning on others. Farebrother et al. (2018) and Rusu et al. (2022) have both previously investigated the transfer and generalization capabilities of value based agents using modes and difficulties of Atari 2600 games. Our work goes further by looking at the impact of pretraining on multiple variants to improve generalization.

## 3 EMPIRICAL METHODOLOGY

**Environment Protocol**: We follow the latest ALE recommendations proposed by (Machado et al., 2018). We use sticky actions to introduce stochasticity in the environment, with probability 0.25 the environment will not execute the action specified by the agent and will repeat the previous action instead. The agent also does not receive a signal when a life is lost within the game. Even when not all actions are available, agents must act with the full set of 18 possible actions. We use the default pre-processing (Mnih et al., 2015) for environment frames.

**ALE Games & Variants**: We will be using four Atari games, SPACE INVADERS, ASTEROIDS, BREAKOUT and BANK HEIST. We chose them because of their relatively large number of game *variants*, 32 for SPACE INVADERS, 66 for ASTEROIDS, 24 for BREAKOUT and 32 for BANK HEIST (see the Appendix for a detailed overview of each game). For each game we split the available *variants* into a train and a test set, where the test set contains 3 unseen *variants*. This split is done randomly, except for SPACE INVADERS where we reuse the test set introduced by Farebrother et al. (2018).

**Pre-training Agent**: All our experiments use IMPALA (Espeholt et al., 2018), a model-free actor-critic method. IMPALA introduces V-traces, an off-policy learning algorithm that corrects updates that become off-policy due to the lag between actors and a learner in a distributed setup. Prior work using IMPALA in the multi-task regime always used an separate head of value function for each task. Here, we use *a single policy and value network* to play every flavour of a game. By restricting the different tasks variants of the same game we observe less negative interference between tasks which allows us to share parameters between the policy and value function. For experiments that analyze the impact of network capacity, we also use deeper ResNets (He et al., 2016) than those used by Espeholt et al. (2018). Further details regarding hyperparameters and the training procedure may be found in the Appendix.

**Reporting & Evaluation**: Given that human scores are not available for Atari 2600 game modes and difficulties, we cannot report human-normalized scores. Instead, we provide results using an *IMPALA-Normalized* score, where we assign a normalized score of 1 to the average score obtained by an IMPALA agent trained for 200 million frames and a normalized score of 0 to the random agent. For reliable evaluation, we report aggregate metrics and performance profiles, with 90% stratified bootstrap confidence intervals (Agarwal et al., 2021b). In particular, we use the interquartile mean (IQM) recommended by Agarwal et al. (2021b), which computes the mean of normalized scores over the middle 50% of the runs, combined across all the test *variants*. IQM is more robust to outliers than mean and results in smaller CIs than median. We report learning curves collected by a separate evaluation process during.

**Pre-training procedure:** Unless otherwise specified, the amount of pretraining data is kept fixed for each agent at 15 billion environment frames. As a result, each agent interacts with each mode and difficulty for at least 200 million frames. The different tasks are split among multiple actors and the agent must play all variants of a game in the training set at the same time. The agent is not provided the mode and difficulty identities with the observation. Though passing this information may improve training performance, it would negatively impact generalization to unseen *variants*.

**Fine-tuning on unseen *variants***: A common way to use a learned representation is to reuse its parameters as initialization to solve a subsequent task. We follow this procedure and use pretrained models to initialize the policy and value network of a new IMPALA agent to play the test set games. Previous work has usually limited the length of the fine-tuning process to 10 million frames, here we choose the standard limit of 200 million frames to quantify the impact of the learned initialization using the standard amount of online interactions used by ALE agents (Machado et al., 2018). Every fine-tuning experiment is run using 10 seeds.

## 4 EXPERIMENTS

Using the above methodology, we now present our results on multi-task pretraining and generalization using the game variants in ALE.

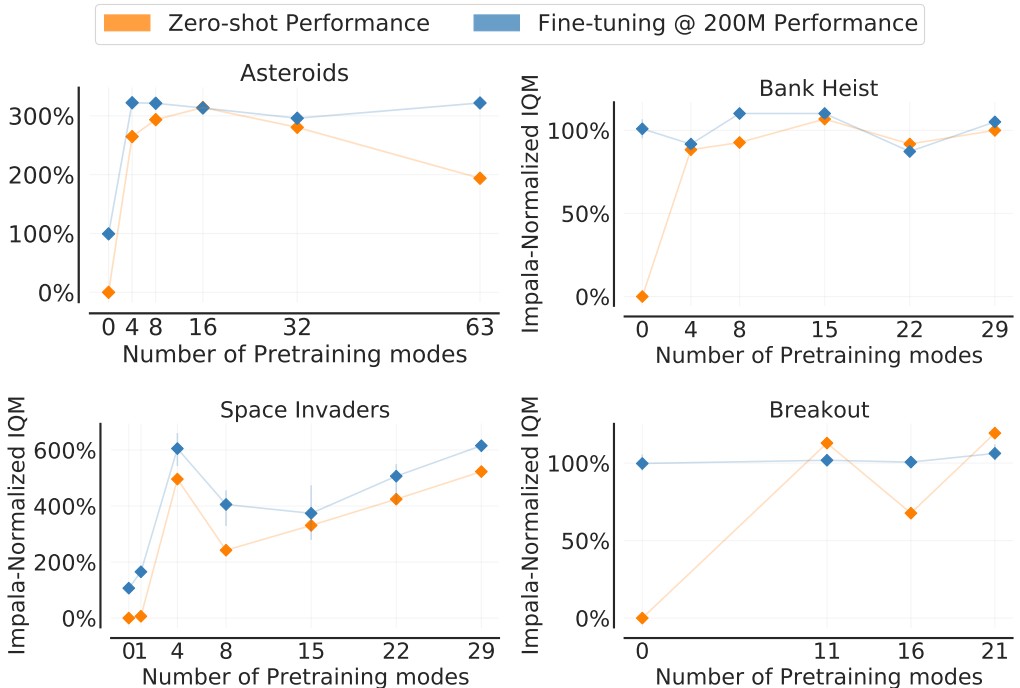

Figure 2: **Zero-shot** *vs.* **Fine-tuning** performance on unseen modes using interquartile mean (IQM) scores. Error bars show 90% bootstrap confidence intervals. Notably, increasing the number of modes for pre-training consistently improves zero-shot and fine-tuning performance after 200M frames of online interactions for most games. These results indicate that the pre-trained representations learn relevant features that can transfer to variants of the same game.

## 4.1 PRETRAINING ON MULTIPLE VARIANTS

RL agents are often trained starting from a randomly initialized policy, as it is unclear whether they can leverage their experience during pretraining to generalize well on subsequent tasks. To understand this behavior we are interested in quantifying the impact of pretraining and number of pretraining variants on the transfer and generalization capabilities of an IMPALA agent (Espeholt et al., 2018). We first compare agents pretrained on subsets containing different numbers of variants of a game using the same budget of 15B environment frames. We evaluate zero-shot transfer by using a pretrained policy directly on a target variant. We also analyze fine-tuning performance by allowing the pretrained policy to interact with the test variant for 200 million environment frames. Given the large number of variants for selected games, it precludes pretraining on all possible subsets of variants as this number grows exponentially with the number of variants. Instead, we randomly subsample multiple subsets of different sizes and require that each subset be a superset of previously picked smaller subsets. This was necessary to limit the effect of particular variants that might have a larger impact on generalization due to their higher similarity with unseen variants (Rusu et al., 2022).

### 4.1.1 ZERO-SHOT

Zero-shot generalization is the most desirable but also most difficult kind of generalization to achieve as it does not allow for any interaction with the target environment. In the context of RL generalization benchmarks with procedurally generated environments, such as Procgen (Cobbe et al., 2020a), zero-shot transfer only emerges after interacting with hundreds of training environments. Here, we assess the zero-shot performance of pretrained models on the test game variants after pretraining on a relatively much smaller number of environments. Results are shown in Figure 2 (orange). Pretraining is highly beneficial on all games – on ASTERIOIDS and SPACE INVADERS pretrained agents achieve a higher score than the non-pretrained IMPALA baseline. We also observe that policies pretrained with more modes generally perform better and benefit from the additional diversity in the pretraining data. While our results contrast with Farebrother et al. (2018), who did not observe much zero-shot

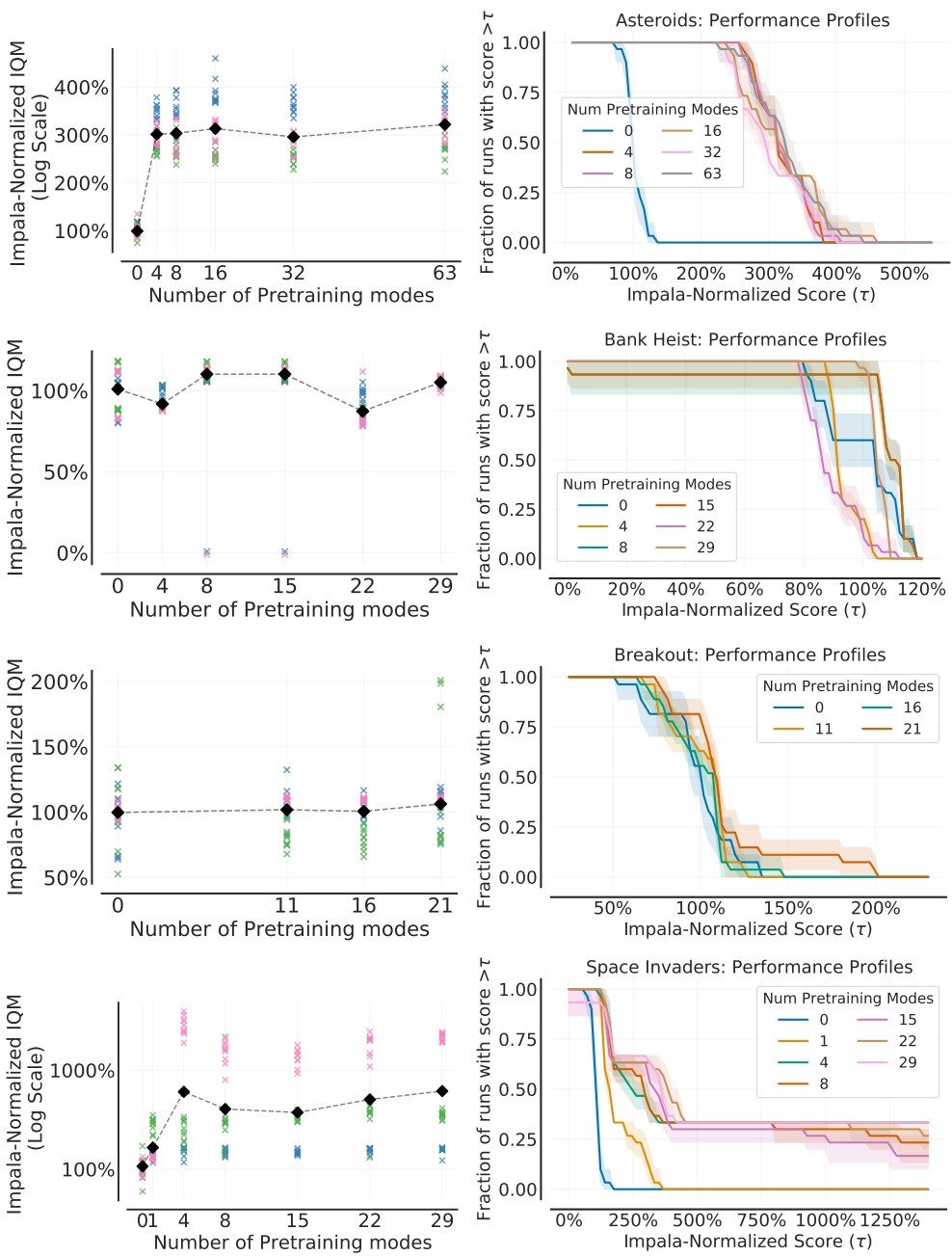

Figure 3: **Fine-tuning performance on unseen modes** after 200M frames. On all games except BREAKOUT, pretrained policies significantly outperform the baseline that was randomly initialized (0 modes). On BREAKOUT, we observe a huge boost in sample efficiency from pretraining but final performance saturates for all agents. **Left**. Aggregate IQM as a function of pretraining modes. Crosses show performance on the 3 individual variants, where each color corresponds to a specific variant. **Right**. Performance profiles are quite useful for qualitative summarization: Higher is better. Area under the profiles correspond to the mean performance, the median can be read by the intersection of the profile curves with the horizontal line at $y = 0.5$. Similarly, we can easily check how many runs score above a certain score threshold $\tau$ using the intersections of the curves with the vertical line $x = \tau$. Shaded region shows 90% stratified bootstrap CIs.

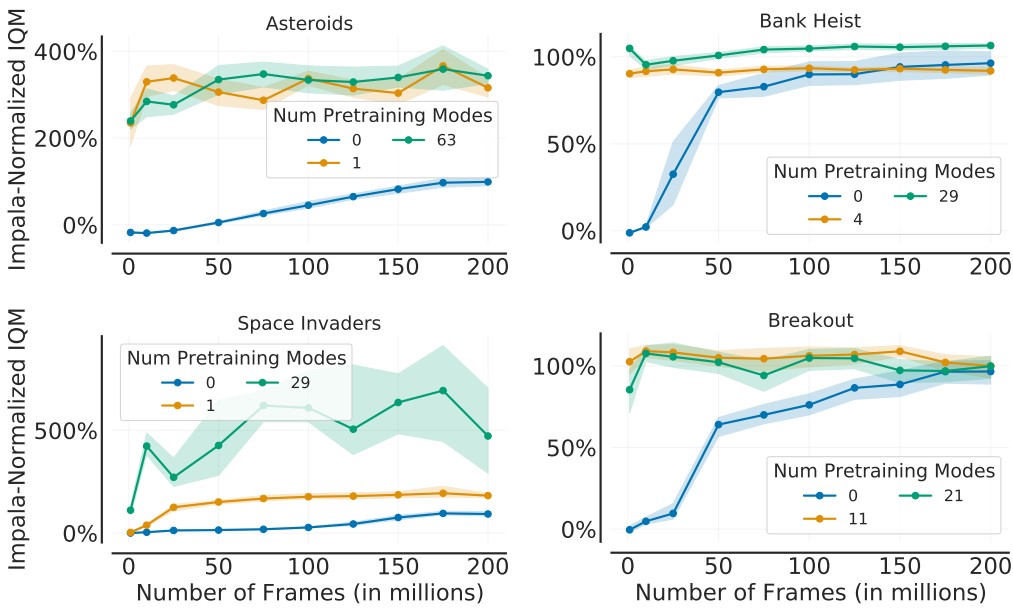

Figure 4: **Sample efficiency** IQM evaluation curves for IMPALA with no pretraining (0 modes), a small number of pretraining modes and a large number of pretraining modes. Shaded regions show 90% bootstrap confidence intervals. Pretrained agents on a large number of modes achieve a much score than agents trained from scratch, in some cases like BREAKOUT and BANK HEIST they do not need to be finetuned.

transfer on Atari after pretraining on a single mode, the two results do not contradict each other as we see that SPACE INVADERS pretrained using a single mode does not generalize well zero-shot. Surprisingly, we find that pretraining on just 4 modes enables good zero-shot transfer.

### 4.1.2   FINE-TUNING

A common approach for transfer is to fine-tune a pretrained model on the test variant, giving time for the policy to adapt to the new task. In this section we focus on quantifying the performance of fine-tuning models after pretraining with different numbers of variants. We report results in Figure 3 and 4. We observe that fine-tuning is valuable for games such as SPACE INVADERS and ASTEROIDS where some models had not achieved the highest score they could attain. On ASTEROIDS and SPACE INVADERS, fine-tuned models significantly outperform the IMPALA baseline trained from scratch and reach 3× - 6× higher scores.

Our result corroborates previous results demonstrating the benefits of pretraining and fine-tuning, in particular, with domain randomization (Tobin et al., 2017; Cobbe et al., 2020b). What is surprising is that, in our setup, these effects are largely captured with a relatively small number of pretraining modes and difficulties, possibly due to the fact that these variants may be sufficient to cover most factors of variations present in the test set. We see little additional improvements when increasing the number of pretraining modes. On the other hand, domain randomization will often generate hundreds of thousands of variations of an environment; Cobbe et al. (2020b)'s experiments required thousands of levels to see an improvement in terms of zero-shot generalization. These results indicate that few-shot transfer is much more difficult to achieve and requires a large amount of diversity in the pretraining data. Nonetheless, representations learned with a handful of modes result in sample-efficient fine-tuning even when they have limited zero-shot transfer capabilities. For example, despite not performing better than a random agent in zero shot transfer, the agent pretrained on a single SPACE INVADERS see its performance increasing rapidly in the first 25 million frames after adapting its representation to the new task at hand. Somewhat surprisingly, the pretrained agents still significantly outperform the same agent trained from scratch (0 modes) even after 200 million frames, whereas we could have imagined that the effect of initialization had been washed away by now. This is in line with recent work (Igl et al., 2020; Nikishin et al., 2022) highlighting the lasting effects of initialization in deep RL.

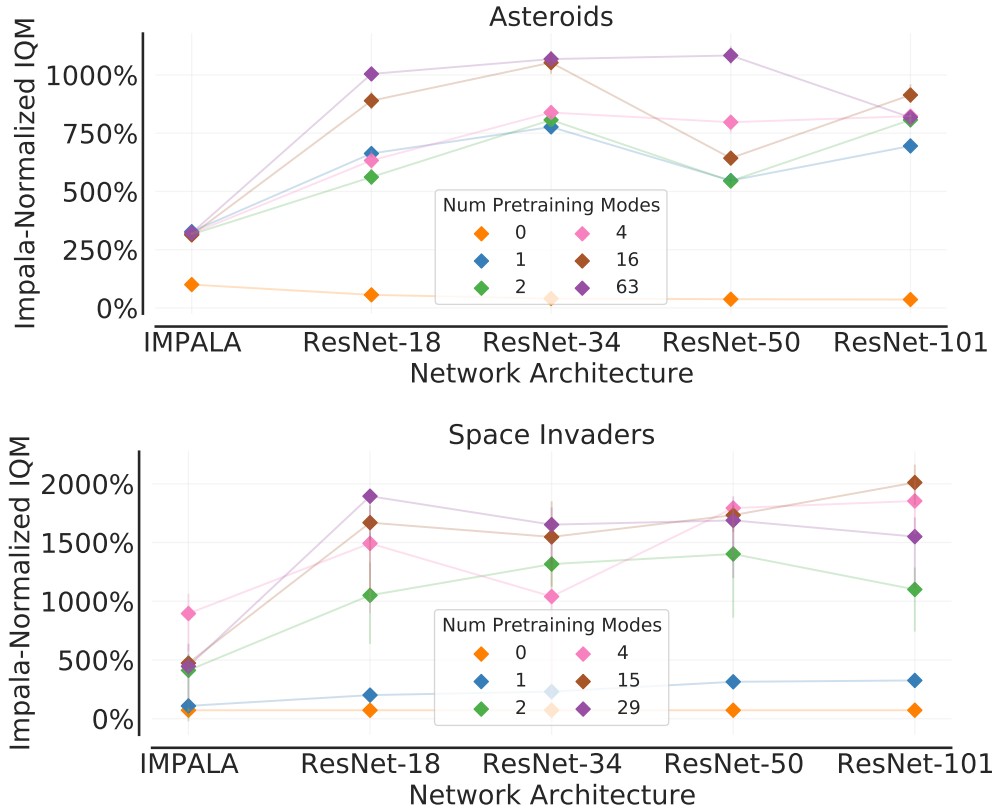

Figure 5: **Impact of network capacity** on fine-tuning performance on unseen modes. 0 modes corresponds to no pretraining. When networks are randomly initialized, ResNets underperform the IMPALA architecture and performance decreases as the network capacity is increased. With pretraining ResNets significantly outperform the IMPALA architecture.

## 4.2 SCALING MODEL SIZE

Our results so far hold for the network introduced with the IMPALA algorithm (Espeholt et al., 2018). Though it is an improvement over the DQN network architecture, the IMPALA network has only 15 layers and less than 2 million parameters. This is relatively small in comparison with models used in vision and NLP where large networks trained on large datasets have been shown to be superior and display impressive generalization capabilities. Additionally, larger capacity models in single-task deep RL settings often lead to instabilities or performance degradation (Van Hasselt et al., 2018; Ota et al., 2021; Bjorck et al., 2021), making it unclear whether multi-task RL can leverage such high-capacity models. We may wonder if increasing the network size is beneficial in the multi-task setting, where there is diversity in the training data and hence more to learn.

In this section we investigate this idea and take a closer look at the generalization capabilities of the IMPALA algorithm combined with larger networks from the ResNet family (He et al., 2016). In our experiments we use ResNet-18, ResNet-34, ResNet-50 and ResNet-101 (respectively 11.4, 21.5, 23.9 and 42.8 million parameters). As before, we pretrain these architectures for 15B frames with varying numbers of game variants before fine-tuning on the test flavours for 200 million frames. Due to the huge computational cost of these experiments we limit our results to ASTEROIDS and SPACE INVADERS. Results are displayed in Figure 5. When training only on test environments, increasing the network size noticeably degrades performance in both games. This result is aligned with prior findings and may explain the dearth of reported results using very large networks in the RL literature. Inversely, pretraining on multiple variants unlocks the potential of larger networks in deep RL. As such, we observe a significant increase in performance (note the logarithmic scale) and these architectures achieve a score 3-4× higher than the IMPALA network trained in the same conditions. Moreover, most of these benefits are attained with ResNet-18, with small benefits in performance

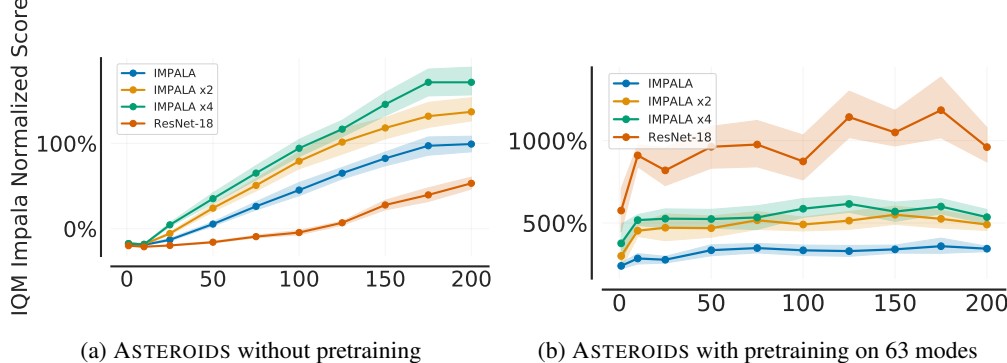

(a) ASTEROIDS without pretraining        (b) ASTEROIDS with pretraining on 63 modes

Figure 6: **Sample efficiency** IQM evaluation curves for IMPALA with no pretraining (0 modes) and a large number of pretraining modes. Shaded regions show 90% bootstrap confidence intervals. IMPALA ×2 and IMPALA ×4 achieve higher returns with and without pretraining. However they remain far below the performance of fine-tuned residual networks showing that these networks generalize better.

when we further increase model capacity. Altogether, it is unexpected to see that pretraining can have such a dramatic impact on the performance of larger networks. In recent years, researchers have attempted to train multi-task policies on all Atari games using slight variations of the IMPALA architecture (Hessel et al., 2019; Guo et al., 2020). Our results indicate that these approaches would likely benefit from higher capacity networks.

## 4.3    SCALING IN WIDTH

Prior work has looked into scaling IMPALA using wider networks by increasing the number of channels in convolutional layers. Cobbe et al. (2020a) multiplied the number of channels in the IMPALA architecture by a factor 2 and 4, they observed better sample efficiency and generalization with wider networks. In this section we attempt to understand how scaling the network in width compares with scaling in depth with residual networks. We evaluate the wider architectures used by Cobbe et al. (2020a), referred as IMPALA ×2 and IMPALA ×4 (respectively 2.4 and 5.5 million parameters) with the baseline IMPALA architecture and ResNet-18. Figures 6 and 9 show learning curves for these architectures on ASTEROIDS and SPACE INVADERS test variants when networks are trained from scratch and after pretraining. We observe that IMPALA ×2 and IMPALA ×4 provide meaningful benefits over the IMPALA baseline. These models achieve higher returns when they are trained from scratch and after pretraining which is line with Cobbe et al. (2020a)'s results. Nonetheless after pretraining their performance still remains far ResNet-18 which achieves almost twice their performance.

## 5    CONCLUSION

Deep reinforcement learning has made a lot of strides in recent years, yet real world applications remain seldom. Better generalization in reinforcement learning could be one of the missing pieces to unlock real-world reinforcement learning. Having access to generalist agents that can perform multiple tasks and are capable of leveraging their prior knowledge to learn new tasks efficiently would take RL to new heights. Nevertheless, prior work has shown that RL remains far from achieving such ambitions and that generalization in RL should not be relied upon. Yet our work offers a glimmer of hope: in our controlled setup, RL agents pre-trained on a small number of game variants generalize to *never-before-seen* variants and considerably improve sample efficiency. Overall, our work calls for using more diverse data and bigger networks in reinforcement learning, which is now a given in other machine learning communities. Though there remains a lot of work to be done to fully take advantage of these two elements, our work offers a glimpse at the potential of large scale multi-task RL. Finally, in line with Agarwal et al. (2022), we would open-source our pretrained models to allow researchers to reproduce our findings as well as study zero-shot generalization and fine-tuning in RL, without the excessive cost of multi-task pretraining from scratch.

## 6 ACKNOWLEDGEMENTS

The authors would like to thank Robert Dadashi, Andrei Rusu and anonymous ICLR's reviewers for providing valuable feedback on an earlier draft of this work.

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

# A    RELATED WORK

**Pretraining in reinforcement learning:** The most common workflow in reinforcement learning is to learn *tabula rasa* by training a randomly initialized agent. However, this is often computationally inefficient, it requires many interactions between the learning agent and the environment and does not make use of knowledge that may be available. The hope of pretraining is to leverage large amounts of training data to accelerate learning on downstream tasks. Recently reinforcement learning practitioners have started to adopt the *pretrain then finetune* paradigm that has become prevalent in other domains like vision (He et al., 2020; Chen et al., 2020) or NLP (Brown et al., 2020; Devlin et al., 2018). Due to the large number of reinforcement learning tasks that rely on perception, many works have advocated for using visual representations pretrained on images or videos. A policy or value function is then learned on top the pretrained representation (Sermanet et al., 2018; Shah & Kumar, 2021; Parisi et al., 2022; Seo et al., 2022; Träuble et al., 2022). Another common strategy for pretraining in reinforcement learning is to use an offline dataset of human or agent interactions to learn a policy using offline RL (Levine et al., 2020). The pretrained policy is then fine-tuned using online RL (Nair et al., 2020; Lu et al., 2021; Kostrikov et al., 2022). Instead of optimizing a policy directly other works have focused on learning a world model that can later used by an agent to improve its policy in a sample efficient manner (Sekar et al., 2020; Seo et al., 2022). Another approach is to learn task-agnostic features through interactions with an environment in the absence of a reward function. These features can be reused in the fine-tuning stage when the reward function becomes available to improve sample efficiency (Eysenbach et al., 2018; Schwarzer et al., 2021; Touati & Ollivier, 2021). For a more detailed overview of the current state of pretraining in RL we refer to Xie et al. (2022)

**Multi task reinforcement learning:** Current successes in RL have been limited to agents trained to solve a single task and designing generalist agents that can solve multiple tasks at the same time remains an open challenge. A popular approach to multi-task reinforcement learning has been to concurrently learn multiple tasks at the same time (Espeholt et al., 2018; Hessel et al., 2019; Guo et al., 2020). Nonetheless, because of negative interference between tasks these models usually perform worse than agents trained on individual tasks. To circumvent those issues previous work have proposed to distill single tasks expert into a multi-task model Parisotto et al. (2015); Rusu et al. (2015); Teh et al. (2017). In the offline setting, recent works have attempted to frame the RL problem as a sequence prediction problem and build on high capacity models based on the Transformer architecture (Vaswani et al., 2017). Gato (Reed et al., 2022) and Multi-Game Decision Transformer (Lee et al., 2022) are multi-task agents trained on expert data that are able to play multiple Atari games. Though one of the ambitions of multi-task reinforcement learning is to take advantage of information from previously learned tasks to learn new tasks faster this is not always observed as negative interference between tasks can impact downstream learning. Yet recent contributions have shown that large scale multi task training on a set of diverse tasks can lead to generalization to new unseen tasks (Kalashnkov et al., 2021; Lee et al., 2022).

# B    TRAINING PROTOCOL

## B.1    HYPERPARAMETERS

In this section, the specific parameter settings that are used throughout our experiments are given in detail. We kept all the original IMPALA hyperparameters fixed except the optimizer where we used Adam (Kingma & Ba, 2014) instead of RMSProp as it was shown to be superior in the context of reinforcement learning (Ceron & Castro, 2021). We did not do any hyperparameter tuning.

We use an efficient implementation of IMPALA (Espeholt et al., 2019; Hessel et al., 2021) that runs on TPUs. Experiment specific hyperparameters are as follow

- The pretraining phase is carried on 32 TPU v3 cores using 6400 actors.

- Fine-tuning is done on 8 TPU v2 cores using 900 actors.

Table 1: Hyperparameters for Atari experiments.

| Parameter | Value |
|---|---|
| Image Width | 84 |
| Image Height | 84 |
| Grayscaling | Yes |
| Action Repetitions | 4 |
| Max-pool over last N action repeat frames | 2 |
| Frame Stacking | 4 |
| End of episode when life lost | No |
| Sticky actions | Yes |
| Action set | Full (18 actions) |
| Reward Clipping | [-1, 1] |
| Unroll Length ($n$) | 19 |
| Batch size | 128 |
| Discount ($\gamma$) | 0.99 |
| Baseline loss scaling | 0.5 |
| Entropy Regularizer | 0.01 |
| Adam $\beta_1$ | 0.9 |
| Adam $\beta_2$ | 0.999 |
| Adam $\epsilon$ | 1e-8 |
| Learning rate | 3e-4 |
| Clip global gradient norm | 40 |

## B.2 ATARI VARIANTS: GAME MODES AND DIFFICULTIES

We briefly review modes and difficulties of Atari games used throughout the main paper. We refer to the original manuals [1] for more in depth information. Rusu et al. (2022) also provides a brief overview of game modes and difficulties for the games BREAKOUT and SPACE INVADERS. To identify individual flavours we use the notation $mXdY$ where $X$ refers to the game mode setting and Y to the difficulty setting:

- **Space Invaders** is a shooting game where the player moves a laser cannon across the bottom of the screen and fires at aliens. The aliens begin on the upper half of the screen and move left and right as a group. The players wins by eliminating all the aliens. There are two difficulty switches and sixteen different game modes leading to a total of 32 flavours. We keep $m1d0$, $m1d1$ and $m9d0$ for the test set and keep the other modes for training.

- **Breakout** is a game where rows of bricks are layered at the top half of the screen. Using a single ball the player must hit the bricks using the paddle it controls at the bottom of the screen. However if the player let the ball fall below the bottom of the screen it looses a life. There are two difficulty switches and twelve game modes for a total of 24 flavours. We use $m12d0$, $m28d1$, $m36d1$ for the test set and use the remaining modes for training.

- **Asteroids**, in this game the player controls a ship in space that must destroy asteroids. The ship must avoid getting hit by asteroids, saucers will also appear regularly and try to shoot the ship. There are two difficulty switches and 33 game modes for a total of 63 flavours. $m4d0$, $m18d3$, $m27d3$ are saved for the test set while the remaining modes are part of the training set.

- **Bank Heist** is a maze game similar to Pac-Man where the objective is to rob as many banks as possible while avoiding police cars that appear everytime a bank is robbed. For this game there are four difficulty switches and eight game modes for a total of 32 different flavours. The training set contains all the flavours except $m8d3$, $m20d1$, $m24d0$ that are used for the test set.

---

[1] https://atariage.com/system_items.php?SystemID=2600&itemTypeID=MANUAL

### B.3 SETS OF VARIANTS USED FOR PRETRAINING

We use the following subsets of modes during the pretraining phase. For each game we randomly subsampled multiple subsets of different sizes and required that each subset be a superset of previously picked smaller subsets to limit the effect of particular variants that might have a larger impact on generalization.

**Asteroids:** We used the following modes to pretrain on ASTEROIDS

- 1 mode: $m9d3$
- 2 modes: $m9d3$, $m17d3$
- 4 modes: $m9d3$, $m10d3$, $m16d0$, $m17d3$
- 8 modes: $m6d3$, $m9d3$, $m10d3$, $m12d0$, $m16d0$, $m17d3$, $m18d0$, $m25d3$
- 16 modes: $m2d3$, $m4d3$, $m5d0$, $m6d3$, $m8d0$, $m9d3$, $m10d3$, $m12d0$, $m16d0$, $m17d3$, $m18d0$, $m20d3$, $m23d0$, $m25d3$, $m28d0$, $m29d3$
- 32 modes: $m0d3$, $m1d0$, $m1d3$, $m2d3$, $m4d3$, $m5d0$, $m6d0$, $m6d3$, $m7d0$, $m7d3$, $m8d0$, $m9d3$, $m10d3$, $m12d0$, $m16d0$, $m16d3$, $m17d3$, $m18d0$, $m20d3$, $m21d0$, $m22d0$, $m23d0$, $m23d3$, $m24d0$, $m25d3$, $m26d0$, $m28d0$, $m29d0$, $m29d3$, $m31d0$, $m31d3$, $m128d0$
- 63 modes: 32 modes + $m0d0$, $m2d0$, $m3d0$, $m3d3$, $m5d3$, $m8d3$, $m9d0$, $m10d0$, $m11d0$, $m11d3$, $m12d3$, $m13d0$, $m13d3$, $m14d0$, $m14d3$, $m15d0$, $m15d3$, $m17d0$, $m19d0$, $m19d3$, $m20d0$, $m21d3$, $m22d3$, $m24d3$, $m25d0$, $m26d3$, $m27d0$, $m28d3$, $m30d0$, $m30d3$, $m128d3$

**Space Invaders:** We used the following modes to pretrain on SPACE INVADERS

- 1 mode: $m5d0$
- 2 modes: $m5d0$, $m12d1$
- 4 modes: $m5d0$, $m8d1$, $m12d1$, $m15d0$
- 8 modes: $m5d0$, $m5d1$, $m6d1$, $m7d1$, $m8d1$, $m10d1$, $m12d1$, $m15d0$
- 15 modes: $m0d1$, $m2d0$, $m3d1$, $m4d0$, $m4d1$, $m5d0$, $m5d1$, $m6d0$, $m6d1$, $m7d1$, $m8d1$, $m10d1$, $m12d1$, $m14d0$, $m15d0$
- 22 modes: $m0d0$, $m0d1$, $m2d0$, $m3d0$, $m3d1$, $m4d0$, $m4d1$, $m5d0$, $m5d1$, $m6d0$, $m6d1$, $m7d1$, $m8d1$, $m9d1$, $m10d1$, $m11d0$, $m12d1$, $m13d1$, $m14d0$, $m14d1$, $m15d0$, $m15d1$
- 29 modes: $m0d0$, $m0d1$, $m2d0$, $m2d1$, $m3d0$, $m3d1$, $m4d0$, $m4d1$, $m5d0$, $m5d1$, $m6d0$, $m6d1$, $m7d0$, $m7d1$, $m8d0$, $m8d1$, $m9d1$, $m10d0$, $m10d1$, $m11d0$, $m11d1$, $m12d0$, $m12d1$, $m13d0$, $m13d1$, $m14d0$, $m14d1$, $m15d0$, $m15d1$

**Bank Heist:** We used the following modes to pretrain on BANK HEIST

- 1 mode: $m8d0$
- 4 modes: $m8d1$, $m12d2$, $m20d0$, $m28d1$
- 8 modes: $m0d2$, $m0d3$, $m4d2$, $m8d1$, $m8d2$, $m12d2$, $m20d0$, $m28d1$
- 15 modes: $m0d0$, $m0d2$, $m0d3$, $m4d2$, $m4d3$, $m8d0$, $m8d1$, $m8d2$, $m12d2$, $m16d2$, $m20d0$, $m20d2$, $m24d1$, $m24d2$, $m28d1$
- 22 modes: $m0d0$, $m0d1$, $m0d2$, $m0d3$, $m4d0$, $m4d2$, $m4d3$, $m8d0$, $m8d1$, $m8d2$, $m12d1$, $m12d2$, $m12d3$, $m16d1$, $m16d2$, $m16d3$, $m20d0$, $m20d2$, $m24d1$, $m24d2$, $m24d3$, $m28d1$
- 29 modes: $m0d0$, $m0d1$, $m0d2$, $m0d3$, $m4d0$, $m4d1$, $m4d2$, $m4d3$, $m8d0$, $m8d1$, $m8d2$, $m12d0$, $m12d1$, $m12d2$, $m12d3$, $m16d0$, $m16d1$, $m16d2$, $m16d3$, $m20d0$, $m20d2$, $m20d3$, $m24d1$, $m24d2$, $m24d3$, $m28d0$, $m28d1$, $m28d2$, $m28d3$

**Breakout:** We used the following modes to pretrain on BREAKOUT

- 11 modes: $m0d0$, $m4d0$, $m4d1$, $m8d0$, $m8d1$, $m20d0$, $m20d1$, $m32d1$, $m36d0$, $m40d1$, $m44d1$

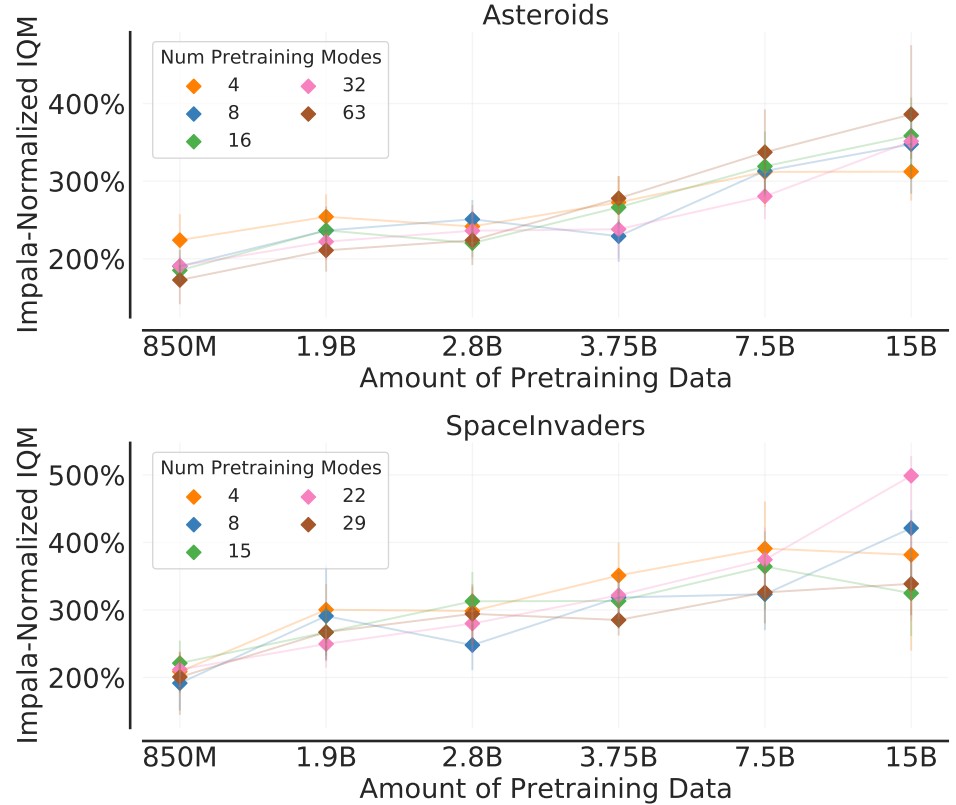

Figure 7: **Impact of pretraining data on fine-tuning performance** on unseen variants. Error bars show 90% bootstrap CIs. Increasing the number of environment interactions during pretraining is beneficial to fine-tuning performance.

- 16 modes: $m0d0$, $m0d1$, $m4d0$, $m4d1$, $m8d0$, $m8d1$, $m16d0$, $m16d1$, $m20d0$, $m20d1$, $m28d0$, $m32d1$, $m36d0$, $m40d0$, $m40d1$, $m44d1$

- 21 modes: $m0d0$, $m0d1$, $m4d0$, $m4d1$, $m8d0$, $m8d1$, $m12d1$, $m16d0$, $m16d1$, $m20d0$, $m20d1$, $m24d0$, $m24d1$, $m28d0$, $m32d0$, $m32d1$, $m36d0$, $m40d0$, $m40d1$, $m44d0$, $m44d1$

## C   ADDITIONAL RESULTS

### C.1   INFLUENCE OF PRETRAINING DATA

Farebrother et al. (2018)'s experiments showed that the limited generalization capabilities of the DQN agent they studied is in part due to an overfitting phenomenon. Their agent is pretrained on a single task and learns a specialized representation for this task that does not generalize well. They show that the zero shot performance of a DQN agent can increase then decrease during pretraining as the representation becomes more specialized. This is highly undesirable and shows how fragile the representation learned by a reinforcement learning agent can be. In this section we investigate whether representations learned using large amounts of pretraining data still overfit to variants encountered during pretraining. To do so we use intermediate checkpoints of the previous pretrained IMPALA models saved at 850M, 1.9B, 2.8B, 3.75B and 7.5B frames. They are then fine-tuned for 200 million frames on the test set variants. Results are shown in Figure 7. We observe that fine-tuning performance improves with more pretraining data and we do not observe the overfitting phenomenon. This is likely due to the fact that joint pretraining on several variants acts as a regularizer on the representation and prevents over specialization. This is akin to how training on large amounts of diverse data from multiple tasks in supervised learning leads to representations that generalize broadly.

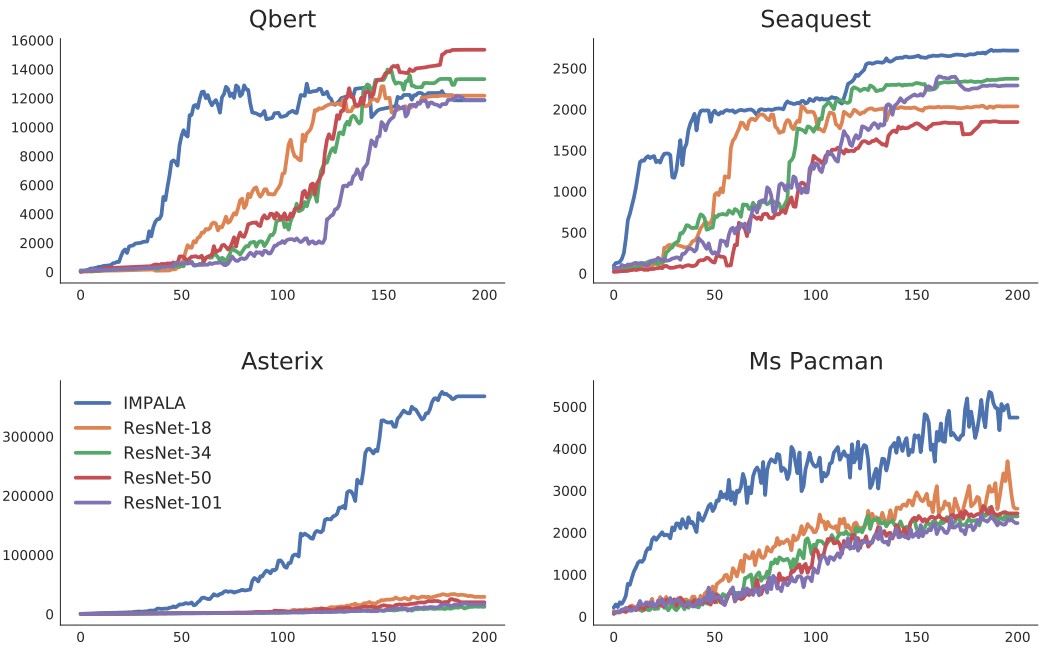

Figure 8: Training curves over 200 million frames on the games QBERT, SEAQUEST, ASTERIX, MS PACMAN with different network architectures. As the model of the model increase the return of the algorithm proportionally decreases. This highlights the difficulty of IMPALA to scale with bigger network architectures.

## C.2 SCALING WITH DEEP NETWORKS

Figure 8 provides additional evidence that the decrease in performance observed using residual networks without pretraining is not limited to ASTEROIDS and SPACE INVADERS. Figure 8 shows evaluation curves on the games QBERT, ASTERIX, SEAQUEST and MS PACMAN with the IMPALA-CNN and ResNets architectures, results are average over 5 seeds. We observe once again that deeper architectures do not benefit from their increased capacity and perform worse than the smaller IMPALA architecture.

## C.3 ADDITIONAL FIGURES

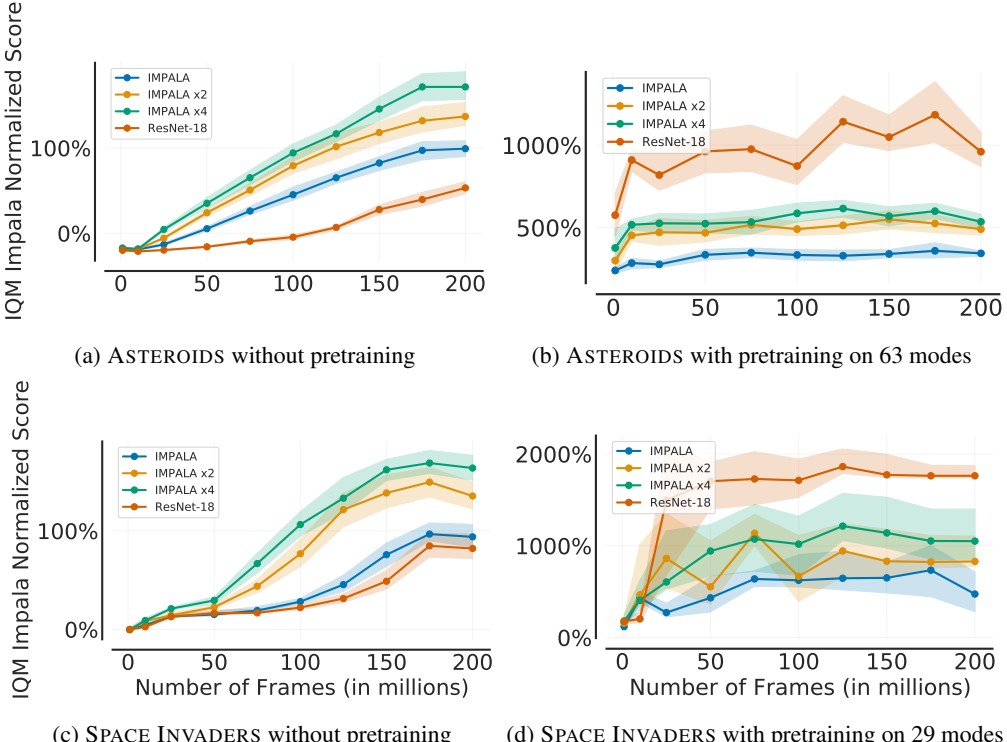

(a) ASTEROIDS without pretraining

(b) ASTEROIDS with pretraining on 63 modes

(c) SPACE INVADERS without pretraining

(d) SPACE INVADERS with pretraining on 29 modes

Figure 9: **Sample efficiency** IQM evaluation curves for IMPALA with no pretraining (0 modes) and a large number of pretraining modes. Shaded regions show 90% bootstrap confidence intervals. IMPALA ×2 and IMPALA ×4 achieve higher returns with and without pretraining. However they remain far below the performance of fine-tuned residual networks showing that these networks generalize better.

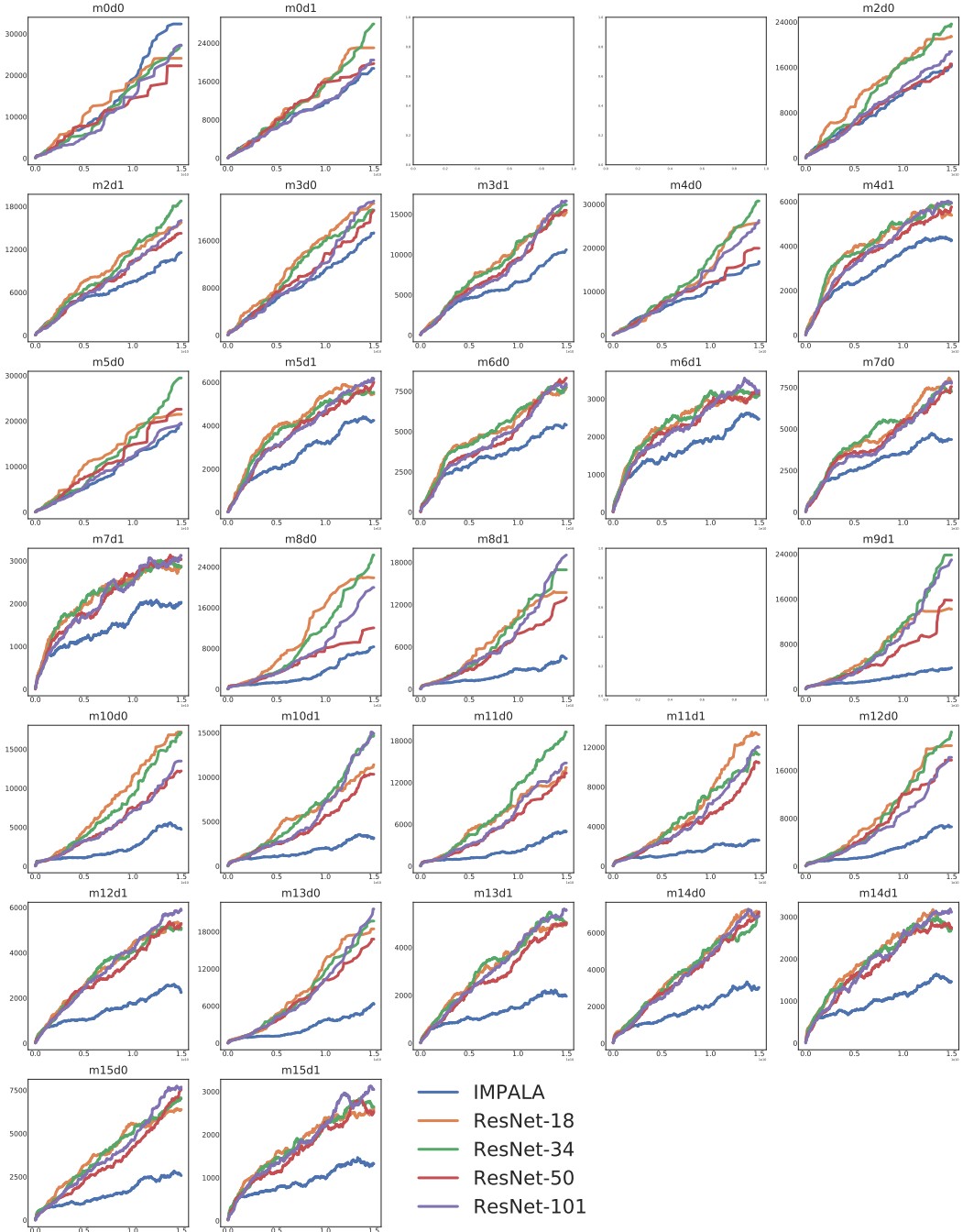

Figure 10: Training curves over 15 billions frames during joint multi-task pretraining on 29 *variants* of the game SPACE INVADERS with different network architectures. The IMPALA-CNN architecture does not have enough capacity to do well on different tasks during multi task training, the architecture chooses instead to focus on a single task m0d0 and underperform the remaining tasks. This is likely due to a lack of capacity.

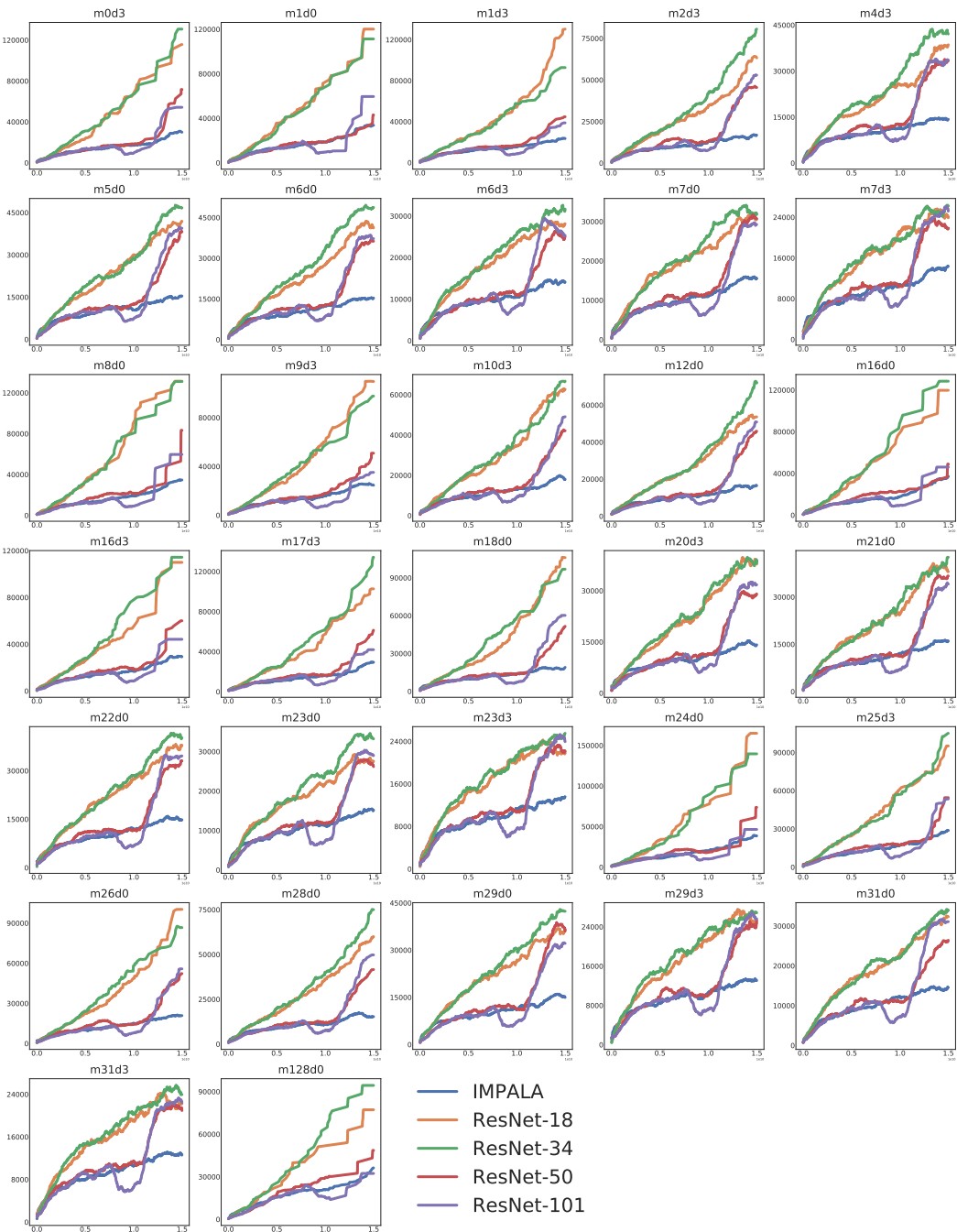

Figure 11: Training curves over 15 billions frames during joint multi-task pretraining on 32 *variants* of the game ASTEROIDS with different network architectures. We observe again that the IMPALA architectures cannot match residual networks in the multi-task regime. Here ResNet-18 and ResNet-34 appear to perform much better than the higher capacity model.

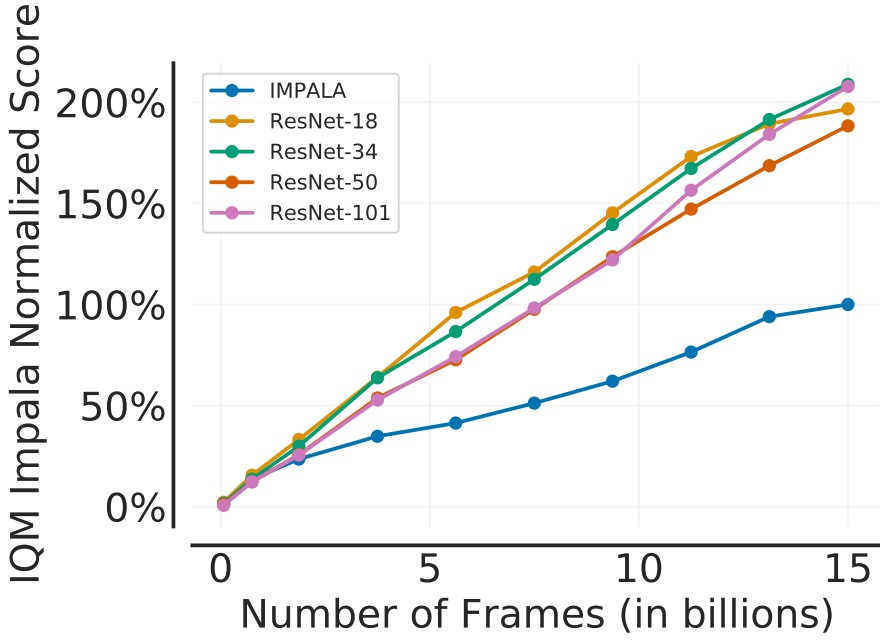

Figure 12: IQM training curves for different network architectures jointly trained for 15 billions frames during multi-task pretraining on 32 *variants* of the game SPACE INVADERS. Normalization for each variant is done using the score achieved by the IMPALA-CNN architecture after multi task pretraining.

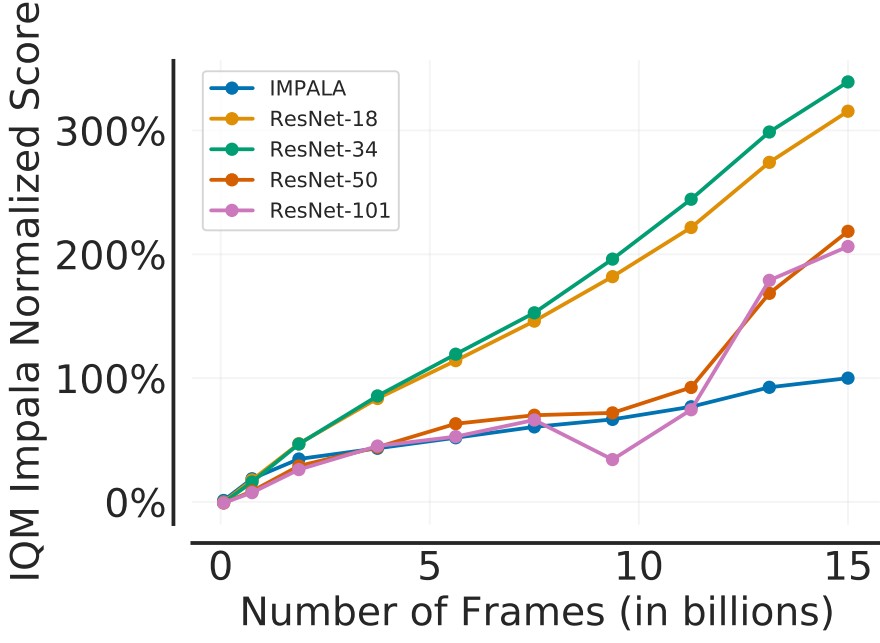

Figure 13: IQM training curves for different network architectures jointly trained for 15 billions frames during multi-task pretraining on 32 *variants* of the game ASTEROIDS. Normalization for each variant is done using the score achieved by the IMPALA-CNN architecture after multi task pretraining.

