# OpenReview forum: "Investigating Multi-task Pretraining and Generalization in Reinforcement Learning"
_ICLR.cc/2023/Conference — ICLR 2023 poster_

### Official Review · Reviewer_9aV5 · 2022-10-20

**Confidence:** 4
**Correctness:** 3
**Technical Novelty And Significance:** 2
**Empirical Novelty And Significance:** 2
**Recommendation:** 5

**Clarity, Quality, Novelty And Reproducibility:**

# Clarity
The paper is written clearly, and the graphs are easy to understand.

# Quality
Overall, the paper appears to be technically sound. The experimental set-up is valid and the analysis of the results is fairly complete. However, the set-up seems to be a bit limited; only the IMPALA algorithm is studied. It would also be interesting to study how similar the tasks have to be for multi-task pre-training to be useful. For example, could we use variants of two similar games such as Pong and Tennis be used?

# Novelty
In my opinion this work has moderate novelty at best. There have been many works on generalization in deep RL using similar environments and algorithms; Machado et al. (2018) uses the same environments and Cobbe et al. (2020) uses IMPALA. As stated in the paper, the main difference is that fine-tuning (with the same number of time steps as the usual single environment Atari benchmark) is considered. The conclusion that pre-training helps is not surprising, given that the diversity of environments is restricted. The use of a ResNet for image-based multi-task deep RL is novel, as far as I know.

# Reproducibility
Code is not provided. However, the authors link to the environments/repositories used and list the hyperparameters, so I think reproducibility is not an issue.

**Strength And Weaknesses:**

# Strengths
- The paper is clear and simple to follow. The authors are clear about the set-up and assumptions.
- The experimental set-up is sound. In particular, the same number of time-steps is used for fine-tuning as for training a single environment on the ALE benchmark.
- The paper shows that we can use a ResNet for policy/value function architecture in some cases.

# Weaknesses
- The settings considered in the paper are fairly limited. Only the IMPALA algorithm is considered, and only variants of the same game are pre-trained on.
- The finding that pre-training helps is not surprising, given the fact that only variants of the same game are pre-trained on and other variants are tested on.

**Summary Of The Paper:**

This paper is an empirical study of the generalization abilities of the deep RL algorithm IMPALA pre-trained on multiple tasks. Pre-training is done on variants of a single game from the Atari benchmark and fine-tuning/testing is done on held-out variants. The authors show that by limiting to variants of a single game, negative interference is reduced enough that the pre-training improves sample complexity (and sometimes final performance as well). However, zero-shot performance does not seem to be affected. In addition, the paper suggests that larger networks, such as those from the ResNet family, can be used in the policy/value function in this multi-task pre-training setting.

**Summary Of The Review:**

The paper is well written, the experimental set-up is sound, and the analysis is clear. However, the setting is somewhat limited, and the results for the most part are not significantly novel.

---

> ### Author Response · Authors · 2022-11-15
> **Author response**
>
> We thank the reviewer for their constructive feedback. We add the following comments to address their concerns.
>
> > Pre-training helps is not surprising, given the fact that only variants of the same game are pre-trained on and other variants are tested on.
>
> Indeed this is what we would expect from a good learning algorithm. Yet, [1] found that there was limited zero-shot transfer and generalization after pretraining on a variant and fine-tuning on another one and [2] found generalization would only happen with a large number of variants (sometimes several hundreds).
>
> We also found that capacity is critical to achieve good generalization, higher capacity models transfer significantly better than the small IMPALA-CNN architecture.  [2] had a similar observation using wider versions of the IMPALA-CNN architecture however our study goes further by considering much larger networks (up to 10x more parameters). Notably we found that pretraining is actually necessary to train deep models like ResNets and **to the best of our knowledge we are the first to successfully train large residual networks like ResNet 101 using only online RL objectives**.
>
> > It would be interesting to see if generalization would happen using similar games like Pong and Tennis.
>
> We agree and hope to tackle multi game transfer in the future.
> It would likely be challenging to observe generalization after pretraining only on a single distinct game (pong and tennis both have a small number of variants).
> However recent papers have shown that pretraining on a large number of games can introduce generalization to unseen games in the context of offline reinforcement learning. [3]
>
> [1] Generalization and Regularization in DQN, Farebrother et al. 2018
>
> [2] Leveraging Procedural Generation to Benchmark Reinforcement Learning, Cobbe et al. ICML 2020
>
> [3] Multi-Game Decision Transformers, Lee et al. NeurIPS 2022.

---

> > ### Comment · Reviewer_9aV5 · 2022-11-19
> > **Updated review**
> >
> > Thanks to the authors for responding. After reading the other reviews, I am more positive on this paper since I feel that the differences from prior work were clarified. However, I am keeping the same score since the scope of the experiments is fairly limited.

---

### Official Review · Reviewer_JzdD · 2022-10-21

**Confidence:** 4
**Correctness:** 3
**Technical Novelty And Significance:** 2
**Empirical Novelty And Significance:** 3
**Recommendation:** 8

**Clarity, Quality, Novelty And Reproducibility:**

The paper is mostly clearly written, the evaluation is done is a methodological way, and the paper is of high quality overall. My thoughts on the novelty of the work are in the section above.

**Strength And Weaknesses:**

### Strengths

* The paper is a good methodical study of an effect of the amount of data/diff environments and the model size on the generalisation properties of an RL agent.
* The paper does not invent a flashy algorithm to push some curves higher but does a thorough study of a phenomenon which might be useful for the researchers and practitioners.
* The effect of number of environments/model size on the fine-tuning performance is important and novel, to the best of my knowledge.

### Weaknesses

* Most of the shown results are known in the community. They have been scattered around different works though, and having them in one place is a plus to the paper in review. However, IMPALA paper has already shown a positive effect of model size on the performance, while the ProcGen paper has shown that training on many seeds (similar to many variants in the current work) improves the zero-shot generalisation performance
* The paper positions itself as the multi-task learning paper, and I would expect it to have a more methodological 'related work' section pinpointing the main literature on the topic.

### Requests/Questions

* Could you, please, add a separate 'related work' section to the paper? Before the additional page is allowed, I am fine with having it in the appendix to be moved to the main text afterwards.
* Figure 6 in log-scale looks a bit confusing to me with most of the curves flattened out. Can you, please, replot them without the log scale in the appendix?
* I think that the paper should be more explicit about the previous works showing similar results: IMPALA’s experiment showing that model size positively affects the performance, and procgen’s number of seeds and model scaling experiments. I don’t think that procgen’s model scaling was discussed in the paper at all. I think these should be added somewhere in the introduction.
* What do you think the limitations of your paper are? Can you add a paragraph on these to the paper?
* You mention that prior IMPALA’s multitask implementation uses a separate critic per task. Can you give any pointers to either the paper or specific implementation?
* “By restricting the different tasks variants of the same game we observe less negative interference between tasks”. Can you provide any evidence for this?
* Figure 6 shows quite a big jump from no pretraining to 4-mode pretraining. Can I see the same curve, but for 1,2 and 3 modes? Is there a huge qualitative jump from 1 or 2 modes pretraining?

### Nits

* Why do you include 1 to the gamma range in Section 2?
* You cite Puterman’s book for MDP, but include \gamma to your MDP tuple. In Puterman’s book, \gamma is not included to the MDP definition.
* Can you plot Figure 3 with shared y-axis? I want to see patterns across the games, and, especially, one subplot in log-scale just makes it confusing.



**Summary Of The Paper:**

The paper investigates how model size and pretaining on similar environments/more data affect the zero-shot and fine-tuning generalisation capabilities of RL agents.


**Summary Of The Review:**

I preliminary choose '6: marginally above', but I am ready to bump the score up if my requests are addressed throughout the discussion period.

---

> ### Author Response · Authors · 2022-11-15
> **Author response to Reviewer JzdD**
>
> We thank the reviewer for the feedback. We have updated the paper and you will find our comments below.
>
> > “However, IMPALA paper has already shown a positive effect of model size on the performance”
>
> The IMPALA paper only showed that model size benefits performance when going from the small architecture used by [1] to the bigger architecture presented in the paper that is referred to as IMPALA architecture in our experiments. However, further increasing the depth of the network is usually harmful and leads to decreased performance as we show in the appendix in Figure 8.
>
>
> > “while the ProcGen paper has shown that training on many seeds”
>
> Procgen has shown that training on many seeds is indeed helpful for generalization, however such generalization requires a much larger number of seeds (e.g., it can be several hundreds in the case of Leaper or FruitBot), which is only possible in procedurally generated environments. We consider a more realistic scenario where the number of seeds is smaller, as often only a finite number of variations are typically available for real-world domains (we are limited by the number of available Atari modes designed for humans).  On the other hand [2] found that DQN would not generalize after pretraining on a small number of seeds. Given the following information it wasn’t clear if generalization was achievable using a small number of modes.
> Our manuscript answers that question and we do observe that generalization is achievable with a small number of seeds (less than 10).
>
> > “Could you, please, add a separate 'related work' section to the paper? “
>
> Thank you for this suggestion, we will make those changes in the next revision of the paper that will be available before the end of the rebuttal period.
>
> > “Can you, please, replot them without the log scale in the appendix?”
>
> We updated the curves without the log scale in Figure 6..
>
> > “I don’t think that procgen’s model scaling was discussed in the paper at all”
>
> We added experiments in the appendix that replicate procgen’s model scaling. We also observe that increasing the width of the impala architecture is beneficial and leads to higher performance for pretrained and non pretrained model. However, the performance of these models remains far below pretrained ResNets and we observe diminishing returns when increasing the width of the network. **This highlights the significance of our results, pretrained ResNets achieve a performance twice higher than the IMPALA-CNN variants.**
>
> > “What do you think the limitations of your paper are? Can you add a paragraph on these to the paper?”
>
> We believe that the main strength of our paper is showing that using pretraining can use larger networks like ResNets in deep RL and they lead to significant improvements. One of the limitation of our paper is that we currently do not know how to achieve the same results without pretraining. Finding a way to do so would make many RL agents much more sample efficient which would be a significant result.
>
> > “You mention that prior IMPALA’s multitask implementation uses a separate critic per task. Can you give any pointers to either the paper or specific implementation?”
>
> In [3] in the background section:  “In this paper, we use the task index at training time, for the value estimates used to compute the policy updates, but not at testing time: our algorithm will return a single general policy $\Pi (A|S)$ which is only function of the individual environment’s state S and not conditioned directly on task index i.”
>
> > “Figure 6 shows quite a big jump from no pretraining to 4-mode pretraining. Can I see the same curve, but for 1,2 and 3 modes? Is there a huge qualitative jump from 1 or 2 modes pretraining?”
>
> We added results with pretraining on 1 and 2 modes in Figure 6. We observe that pretraining IMPALA-CNN or ResNets with such a small number of modes still results in improvements over non pretrained models. This is in contrast to results from [3]., which showed that pretrained 3-layer CNN architectures on one mode doesn’t improve over training from scratch.
> However the limited diversity in the pretraining data limits the performance of these models, models pretrained with a larger number of modes perform better.
>
> [1] Human-level control through deep reinforcement learning, Mnih et al. Nature 2015
>
> [2] Generalization and Regularization in DQN, Farebrother et al. 2018
>
> [3] Multi-task Deep Reinforcement Learning with PopArt, Hessel et al AAAI2019

---

> > ### Comment · Reviewer_JzdD · 2022-11-15
> > **response**
> >
> > Thanks a lot for all your improvements and updates to the paper. I have  a couple of questions left.
> >
> > > On the other hand [2] found that DQN would not generalize after pretraining on a small number of seeds. Given the following information it wasn’t clear if generalization was achievable using a small number of modes. Our manuscript answers that question and we do observe that generalization is achievable with a small number of seeds (less than 10).
> >
> > Do you think this might be because of the differences in the benchmarks?
> >
> > > Thank you for this suggestion, we will make those changes in the next revision of the paper that will be available before the end of the rebuttal period.
> >
> > Great, looking forward to it!
> >
> > > We believe that the main strength of our paper is showing that using pretraining can use larger networks like ResNets in deep RL and they lead to significant improvements. One of the limitation of our paper is that we currently do not know how to achieve the same results without pretraining. Finding a way to do so would make many RL agents much more sample efficient which would be a significant result.
> >
> > That's an interesting thought, but it's not the limitation of the current paper, I would say. For instance, one limitation I can see is that the 'small-num-of-seed' pretraining ability to generalize you've demonstrated is due to the difference in the benchmarks and won't be possible with procgen. Can you think of any others?
> >
> > > In [3] in the background section: “In this paper, we use the task index at training time, for the value estimates used to compute the policy updates, but not at testing time: our algorithm will return a single general policy Π(A|S) which is only function of the individual environment’s state S and not conditioned directly on task index i.”
> >
> > I am a bit confused about this. Doesn't the quote says the opposite? It says that values estimates are computed providing the task id to the network which would be useless if used with separate critics?

---

> > > ### Author Response · Authors · 2022-11-15
> > > **Follow-up response**
> > >
> > > > Do you think this might be because of the differences in the benchmarks?
> > >
> > > Indeed, ALE modes were designed to be challenging for humans and are free of researcher bias, while ProcGen was designed for testing RL generalization. In addition to this, we believe there are two reasons for differences in our results from [2]:
> > > - **Exact number of seeds**: [2] only used a single seed for pretraining. We also observed limited zero shot transfer on Space Invaders after pretraining on a single seed but substantial benefit in fine-tuning even after single-seed pretraining. That said, using as few as only two seeds led to better results for this game, as shown in **Figure 6**.
> > > - **Difference in network architecture**: The 3-layer DQN CNN network used by [2] is much smaller than the IMPALA-CNN architecture, our results in **section A.1** in the appendix emphasize that residual networks with more parameters generalize better. Also, Procgen only scaled the width of Impala-CNN, while our results in Figure ?? show that deeper ResNets outperform IMPALA-CNN for pretraining and fine-tuning.
> > >
> > >
> > > > That's an interesting thought, but it's not the limitation of the current paper, I would say. For instance, one limitation I can see is that the 'small-num-of-seed' pretraining ability to generalize you've demonstrated is due to the difference in the benchmarks and won't be possible with procgen. Can you think of any others?
> > >
> > > Due to the computational cost of our experiments we had to make some concrete choices in our experimental setup. All our experiments are run on the ALE, use the same agent, IMPALA, and only a few network architectures. It would be interesting for future work to consider other environments such as continuous control tasks from pixel input, extend our study to value-based agents and consider more recent network architectures.
> > >
> > > Nonetheless making these concrete choices allow us to focus our attention and resources on quantifying the impact of pretraining on generalization in RL and the dependance on the number of variants, amount of pre-training data and network architecture. We also made sure to provide enough seeds to report aggregate metrics for reliable evaluation of our results. Though we expect our findings like the importance of deeper networks like ResNets for generalization to hold in other domains, our work has not demonstrated this. We will add a paragraph in the paper.
> > >
> > >
> > > > I am a bit confused about this. Doesn't the quote says the opposite? It says that values estimates are computed providing the task id to the network which would be useless if used with separate critics?
> > >
> > > The task id is used to specify which task-specific head of the value network is used to compute the value function. This is made more clear in the **Scale invariant updates for multi-task learning** subsection of the paper: “Consider now a multi-task value function $v$(S) with $N$ outputs, one for each task“ and “For each rollout, only the $i$ the head in the value net is updated, while the same policy network is updated irrespectively of the task“. We’ll update our paper to clarify this too.

---

> > > > ### Author Response · Authors · 2022-11-18
> > > > **Follow-up**
> > > >
> > > > Dear Reviewer JzdD
> > > >
> > > > We have added a related work section in the **Appendix, section B**. We hope our rebuttal answered all your concerns, please let us know if you have any other questions.

---

### Official Review · Reviewer_t4n9 · 2022-10-23

**Confidence:** 4
**Correctness:** 4
**Technical Novelty And Significance:** 2
**Empirical Novelty And Significance:** 4
**Recommendation:** 8

**Clarity, Quality, Novelty And Reproducibility:**

Clarity: The paper is easy to read and understand, and all the results are presented clearly.

Novelty: The experiments presented are novel, and produce conclusions which were unknown beforehand.

Quality: I think the work is of high quality, as mentioned in the strengths section.

Reproducibility: Unfortunately, due to the high computational costs of running all the experiments, the results aren't that reproducible without a similar level of compute. However, if the authors released the data from all their experiments, then that would enable others to build on and analyse that data without having to rerun expensive experiments.

**Strength And Weaknesses:**

Strengths: The questions asked and answered by the paper are interesting and unanswered, and the rigorous experiments provide robust conclusions in this setting, which I'd expect to at least somewhat generalise to settings other than Atari. The paper is well-written and easy to follow, and all the results are clearly presented. I think the results are novel and perhaps counter to some prevailing wisdom in the RL community about the appropriate model size and the value of pretraining. The use of fine-tuning to evaluate the impact of pretraining is good, as it shows conclusions that previously we didn't have.

Weaknesses: There are few weaknesses that I can see. Obviously it would be better if the experiments were demonstrated on more environments, perhaps ones which are different from atari games in some way, but I think the current experiments provide sufficient evidence for the conclusions.

Some suggestions to improve the completeness of the paper: I'd like to see the normalised reward curves during pretraining for the various model sizes, to see if increasing model size also improves pre-training performance as well as fine-tuning performance. Also, given the amount of data gathered on both pretraining dataset size and model size, it would be interesting to see if a functional form for transfer performance similar to https://arxiv.org/abs/2102.01293 can be found, which would allow us to answer questions such as "how much pretraining data do we need to reduce the amount of fine-tuning data by 5x while maintaining the same performance?". If the functional form was similar for the different games, then it could provide guidance for how to trade off pretraining and fine-tuning compute.

**Summary Of The Paper:**

This paper analyses an RL training regime consisting of multi-task pretraining on a set of atari game variants followed by fine-tuning on an unseen variant. They show that while multi-task pretraining often doesn't help zero-shot performance (in particular scaling training time and number of variants doesn't affect zero-shot transfer performance), it improves the fine-tuning performance and sample efficiency significantly, and these improvements scale as pretraining dataset size and number of variants increases. They further show that in this multi-task pretraining regime, larger models than previously used can provide strong benefits to both pretraining and fine-tuning performance.

**Summary Of The Review:**

Overall, I think the paper is worthy of acceptance, and hence I'm recommending an 8. I think the paper's experiments are novel and conclusions interesting and in some cases surprising. In general I'm in favour of more empirical scientific work of this kind, as I think it's something the RL community is lacking.

I think it would be difficult for my score to rise from an 8 to a 10, and it would probably require either: similar experiments on another qualitatively different class of environments to make the results even more robust; the discovery of a robust functional form for fine-tuning performance which holds across environments; or experiments for different algorithm types, to see if these results also generalise to Q-learning based and/or model-based methods.

---

> ### Author Response · Authors · 2022-11-15
> **Author response to Reviewer t4n9**
>
> We thank the reviewer for these valuable comments and address some the comments below.
>
> > “I'd like to see the normalised reward curves during pretraining for the various model sizes, to see if increasing model size also improves pre-training performance as well as fine-tuning performance”
>
> We added these curves to the **appendix, see Figure 9 and 10**. We did not normalize the curves due to the cost of running the Impala baseline on every individual variant with several seeds. However our plots clearly show that there is a big gap in performance between the original impala architecture and residual networks during pretraining. This seems to indicate that pretraining performance correlates with fine-tuning performance.
>
> > “Also, given the amount of data gathered on both pretraining dataset size and model size, it would be interesting to see if a functional form for transfer performance similar to https://arxiv.org/abs/2102.01293 can be found”
>
> We agree that deriving a similar functional form for transfer performance would be interesting.
> Recent work [1] has attempted to derive similar results for reinforcement learning in the multi-agent setting. For single agent reinforcement learning the task appears more complicated, once the maximum achievable return is achieved what is the quantity that would scale with more data? It would not be the return and cannot be the loss either.
>
> [1] Scaling Laws for a Multi-Agent Reinforcement Learning Model, Neuman and Gros, arxiv preprint 2022

---

> > ### Comment · Reviewer_t4n9 · 2022-11-15
> > **Response**
> >
> > Thanks for adding the results from pretraining, that's useful to see.
> >
> > I think it's acceptable to fit a scaling trend functional form on the part of the X axis where reward isn't maximised, or more generally broken functional forms like those in https://arxiv.org/abs/2210.14891 could be used to model a form with two components (the first showing the scaling up, the second the plateau as reward is maximised). I think Impala-normalised return as the y axis would still be a good start (even though it's not the perfect metric, I agree).

---

> > > ### Author Response · Authors · 2022-11-18
> > > **Follow up response to Reviewer t4n9**
> > >
> > > > I think it's acceptable to fit a scaling trend functional form on the part of the X axis where reward isn't maximised, or more generally broken functional forms like those in https://arxiv.org/abs/2210.14891 could be used to model a form with two components (the first showing the scaling up, the second the plateau as reward is maximised)
> > >
> > > We agree this is a good idea though outside of the scope of this work. We will keep it in mind for future follow-up work.
> > >
> > > > I think Impala-normalised return as the y axis would still be a good start (even though it's not the perfect metric, I agree).
> > >
> > > We have added normalized IQM training curves in the **Appendix, Figures 12 and 13)** in the latest revision of the paper. To normalize the curves we use the score achieved after pretraining by the IMPALA baseline trained on multiple variants.

---

### Official Review · Reviewer_eTGE · 2022-10-25

**Confidence:** 4
**Correctness:** 2
**Technical Novelty And Significance:** 1
**Empirical Novelty And Significance:** 1
**Recommendation:** 3

**Clarity, Quality, Novelty And Reproducibility:**

The paper is clearly written although there are some points that need rework:
*Figure 1, the numbers within the images are never explained
* First paragraph section 3, it misses an explanation about what are sticky actions
* The end of that paragraph, explain what is that pre-processing (you can do it in the appendix but then refer it)
* Third paragraph, you mention you restrict the variants of the games, how?
* fourth paragraph, how many seeds you used for Impala normalized scores?
*fifth one, it is not evident why 15 billion frames assures that agents interact with each mode for at least 250 million frames

Since the paper is all about the Atari experiment quality is not good given that many details about reproducibility are missing and that it doesn't contrast the results with existing literature. Still I believe this is fixable in the rebuttal time.

Novelty and significance is the biggest issue here since the contributions of this work are all well known things within the current literature

**Strength And Weaknesses:**

The paper pursues a computationally challenging and ambitious goal. However I believe it fails short to be considered for publication at ICLR since the contributions and outcomes don't really provide any new information about RL or deep learning. The paper also fails to contrast with their results with relevant literature in generalization. Specifically:

* The fact that agents don't do well in the new variants of the task with zero-shot is not surprising since the variant could mean that an optimal policy in the previous variants don't necessarily translate into the new variant. It would have been good to track from which variants the agent was learning and what is the new thing being asked. For instance if I have a variant in the test set that for the first time requires the agent to shoot to air balloons it is more meaningful to evaluate one/two-shot learning to see how the agent performs in the new task. This would explain the contradiction between the results from this work, were authors suggest that DRL is weak at zero-shot generalisation, while there are plenty of examples [1-4] to name a few, of the opposite.

* That pre-training helps to up to certain point is quite known already. For example e.g. ALOE workshop at ICLR 2022 and other workshops mixing unsupervised learning and reinforcement learning have plenty of works on the topic already [5,6] are examples of it

* That larger networks generalize better and that are able to handle multi-task learning better it is also well known [1,7,8]. Specially relevant for this size of networks are transformers architectures, which aren't part of this work either

In general I am afraid that this work is too far below the bar of significance and novelty required for ICLR. One possible direction for this work would strength the study to include different kind of networks, e.g. relational networks, transformers. GNNs... and also use some additional benchmarks different to Atari. Current results suggest that there is a moment that pretraining in those variants is not useful anymore, finding an automatic way to find relevant variants / automatic curriculum may be appealing.

[1] Hill, Felix, et al. "Grounded Language Learning Fast and Slow." International Conference on Learning Representations. 2021.

[2] Hill, Felix, et al. "Environmental drivers of systematicity and generalization in a situated agent." International Conference on Learning Representations. 2020.

[3] León, Borja G., Murray Shanahan, and Francesco Belardinelli. "In a Nutshell, the Human Asked for This: Latent Goals for Following Temporal Specifications." International Conference on Learning Representations. 2021.

[4] Vaezipoor, Pashootan, et al. "Ltl2action: Generalizing ltl instructions for multi-task rl." International Conference on Machine Learning. PMLR, 2021.

[5]Seo, Younggyo, et al. "Reinforcement learning with action-free pre-training from videos." International Conference on Machine Learning. PMLR, 2022.

[6] Yi, Mingyang, et al. "Improved ood generalization via adversarial training and pretraing." International Conference on Machine Learning. PMLR, 2021.

[7] Brown, Tom, et al. "Language models are few-shot learners." Advances in neural information processing systems 33 (2020): 1877-1901.

[8] Chan, Stephanie CY, et al. "Data Distributional Properties Drive Emergent Few-Shot Learning in Transformers." arXiv preprint arXiv:2205.05055 (2022).

**Summary Of The Paper:**

This paper presents an study  on the effect of multi-tasks pretraining in the context of Atari 2600 with deep reinforcement learning agents. It does so by pretraining in multiple variants of 4 games and testing in unseen variants of the game for zero-shot generalization and fine-tuning -for 200 million frames- to see if pre-training helps the agent.

**Summary Of The Review:**

The paper presents a large experiment focused on Impala with varied residual nets Atari games variants. However the outcomes of the experiment are all well abundant in existing literature.

---

> ### Author Response · Authors · 2022-11-15
> **Author response to Reviewer eTGe (part 1)**
>
> We thank the reviewer for their valuable feedback. We have updated our paper with new results and address their comments below.
>
> > “That pre-training helps to up to certain point is quite known already”
>
> While pre-training helps in deep learning, it is not clear whether the same holds for reinforcement learning. On the contrary, prior results [6] showed that RL pre-training on related domains doesn’t lead to performance benefits. Furthermore, another surprising finding is that the benefits of pretraining have long lasting effects. Specifically, after 200M environment – which would correspond to 38 days of  human play – we would think that any benefits of pretraining would have been washed away but pretrained models significantly outperform non pretrained models. This finding has also not been presented in previous literature and highlights the importance of initialization in deep rl.
>
> > “For example e.g. ALOE workshop at ICLR 2022 and other workshops mixing unsupervised learning and reinforcement learning have plenty of works on the topic already [5,6] are examples of it”
>
> We thank the reviewer for providing these references and will add them to the next revision of the manuscript. However we believe that these references consider settings different from ours. [9] uses world models, this introduces a supervised learning task that will indeed benefit from pretraining, [10] does not appear to consider online reinforcement learning and [11] assumes access to a noise-free labelling function to identify high-level domain features which is not available for Atari games. Pretraining a value or policy network is still not common for online reinforcement learning on complex visual environments like Atari games.
>
> > “It would have been good to track from which variants the agent was learning and what is the new thing being asked”
>
> We specified the variants used for pretraining in the appendix. We will also add a section describing the different factors of variations between variants. In the meantime [8] Figure 1 provides a good overview of the variations between game modes and difficulties for Space Invaders and Breakout. Similar information is available for Asteroids (https://atariage.com/manual_html_page.php?SoftwareID=828) and Bank Heist (https://atariage.com/manual_html_page.php?SoftwareLabelID=1008).
>
> > “That larger networks generalize better and that are able to handle multi-task learning better it is also well known”
>
> This has not shown to be true in reinforcement learning. On Atari games, larger networks such as ResNets can degrade performance and underperform the standard IMPALA architecture. To support our claim and show that the issue observed on Figure 6 is not limited to the games Asteroids and Space Invaders, we provide additional learning curves for Qbert, Seaquest, Asterix, and MsPacman without pretraining **in the appendix, Figure 7**. **These curves clearly show that the return decreases as model capacity increase.**
>
> Previous works have observed similar issues when attempting to train larger networks in reinforcement learning [1-2]. It could be argued that we do not benefit from increasing the network size because this may not be necessary for solving a single game. **Yet our results show that we do benefit from more capacity and it can lead to significant improvements even within a single environment, we believe that this is a significant result**. Previous attempts to scale IMPALA like [7]  have focused on scaling in width. However experiments displayed in **Figure 10 in the Appendix** shows that this has limited benefits compared to scaling in depth with deep networks like ResNets.
>
> Multi task learning is also challenging in reinforcement learning. On Atari games, multi-task models that learn to play all games perform significantly worse than policies trained on each game separately [3]. To play distinct Atari games these models usually require a separate critic for each game which limits their generalization power. [4] presents some of the reasons why multi task learning is currently challenging in reinforcement learning.
> **To the best of our knowledge our work shows for the first time that high capacity networks like ResNets are helpful in the multi task setting on complex environments like Atari games.**
>
> > “One possible direction for this work would strength the study to include different kind of networks, e.g. relational networks, transformers. GNNs”
>
> We agree these architectures would be interesting for future work. That said, we did not consider these architectures as none of them are typically used by reinforcement learning researchers on prevalent benchmarks like the ALE. Transformers are notorious for being unstable to train and use in online reinforcement learning [5].

---

> > ### Author Response · Authors · 2022-11-15
> > **Author response (part 2)**
> >
> > >  “it misses an explanation about what are sticky actions”
> >
> > We added an explanation of sticky actions in the environment protocol.
> >
> > >  “how many seeds you used for Impala normalized scores.”
> >
> > We used 10 seeds for all fine-tuning experiments in the new revision of the paper, 5 were used previously.
> >
> > > “ it is not evident 15 billion frames assures that agents interact with each mode for at least 250 million frames”
> >
> > The actual correct answer is 238 million and has been updated in the paper. IMPALA does not use a prioritized replay buffer, because of that it interacts with each variant for the same number of frames during the pretraining phase. The maximum number of modes used for pretraining is 63 for Asteroids, in that case IMPALA would interact with each variant for 15B / 63 = 238 million frames. The remaining experiments use 32 variants or less and in that case IMPALA interacts with each variant for at least 460 million frames.
> >
> > > “you mention you restrict the variants of the games, how?”
> >
> > We only make sure that for a given game, larger subsets of games used for pretraining are supersets of smaller subsets. This is done to limit the impact that some variants could have on generalization if we were to use disjoint sets. We clarify this in section B.2 in the appendix.
> >
> > > “the paper is all about the Atari experiment quality is not good given that many details about reproducibility are missing and that it doesn't contrast the results with existing literature”
> >
> > Now that we specified the subset of variants used for pretraining we believe that our experiments should be easily reproducible
> > We use a proven learning algorithms with implementations available in the most popular deep learning frameworks such as [jax](https://github.com/deepmind/acme/tree/master/acme/agents/jax/impala), [pytorch](https://github.com/facebookresearch/torchbeast) or [tensorflow](https://github.com/deepmind/scalable_agent).
> > The hyperparameters used are specified in the appendix.
> > The environments used in the paper are also all open-sourced.
> >
> >
> >
> > [1] Towards Deeper Reinforcement learning with Spectral Normalization, Bjorck et. NeurIPS 2021
> >
> > [2] Training Larger Networks for Deep Reinforcement Learning, Ota et al.
> >
> > [3] IMPALA: Scalable Distributed Deep-RL with Importance Weighted Actor-Learner Architectures, Espeholt et al., ICML 2018.
> >
> > [4] Ray Interference: a Source of Plateaus in Deep Reinforcement Learning, Schaul et al.
> >
> > [5] Stabilizing Transformers for Reinforcement Learning, Parisotto et al. 2019
> >
> > [6] Generalization and Regularization in DQN, Farebrother et al. RLDM 2019
> >
> > [7] Leveraging Procedural Generation to Benchmark Reinforcement Learning, Cobbe et al. ICML 2020
> >
> > [8] Probing Transfer in Deep Reinforcement Learning without Task Engineering, Rusu et al. CoLLa 2022
> >
> > [9] Seo, Younggyo, et al. "Reinforcement learning with action-free pre-training from videos." International Conference on Machine Learning. PMLR, 2022.
> >
> > [10] Yi, Mingyang, et al. "Improved ood generalization via adversarial training and pretraing." International Conference on Machine Learning. PMLR, 2021.
> >
> > [11] Vaezipoor, Pashootan, et al. "Ltl2action: Generalizing ltl instructions for multi-task rl." International Conference on Machine Learning. PMLR, 2021.

---

> > > ### Author Response · Authors · 2022-11-18
> > > **Follow up**
> > >
> > > Dear Reviewer eTGE
> > >
> > > We hope that our rebuttal provided satisfying answers your concerns. If you have additional questions please let us know.

---

> > > > ### Comment · Reviewer_eTGE · 2022-11-20
> > > > **Response**
> > > >
> > > > I want to thank the authors for their detailed response and also for updating the paper with the various clarifications we asked about. I is shows indeed that authors are doing an intense work here in a challenging and relevant line of work. However, I still humbly believe this paper needs substantial rework before being published for the reasons I stated in my original review.
> > > > Following up with your response:
> > > >
> > > > * "Furthermore, another surprising finding is that the benefits of pretraining have long lasting effects" I am afraid this is not surprising but quite natural instead. Erhan et al. 2010 provide good examples of why pretraining can change the whole learning trajectory of a machine learning model. Authors also suggest that this could be a first indication that pretraining helps in RL and Atari, but I would like to point out to the literature referred in Zhihui, et al. 2022 survey, and also works like Schwarzer et al. 2021 that give evidence on how pretraining helps in Atari.
> > > >
> > > > Authors do refer to some of these related works in the appendix (I strongly believe related work section should be in the main body) but fail to explain how these results are different from previous positive results.
> > > >
> > > > * Regarding the variants, I am afraid that without the promised update (right now it's is a conglomerate of letters and numbers without a mapping to understand them) we cannot assess what is the skill transfer evaluated here. This is important because in RL we cannot divide train and test tasks just randomly since the skill transfer that they may require could be impossible even for a human to do in zero-shot.
> > > >
> > > > * "This has not shown to be true in reinforcement learning. On Atari games, larger networks such as ResNets can degrade performance and underperform the standard IMPALA architecture." I was very surprised to read this statement given the fact that in the original IMPALA paper authors do experiment with a "deep IMPALA" -IMPALA with resnet- and "shallow IMPALA" with a CNN, and deep IMPALA shows much better performance in the Atari games. Since authors of this work had opposite findings it would be very interesting to highlight this contradiction with the original paper and find out what is causing this phenomenon.
> > > >
> > > > *I do agree with the authors that, if we focus on visual encodings only, there is still little research on how going towards deeper and deeper networks can be made beneficial in RL. For this, I believe that future iterations of this work, e.g., with multiple domain experiments and ablations on the representations learnt by the agents, could be quite beneficial for the community.
> > > >
> > > > * As an extra note, transformers were originally unstable but they are already being used effectively in online RL. For instance of the works I mentioned in my original review [1] achieves top performance with a TransformerXL.
> > > >
> > > >
> > > > Erhan, Dumitru, et al. "Why does unsupervised pre-training help deep learning?." Proceedings of the thirteenth international conference on artificial intelligence and statistics. JMLR Workshop and Conference Proceedings, 2010.
> > > >
> > > > Xie, Zhihui, et al. "Pretraining in Deep Reinforcement Learning: A Survey." arXiv preprint arXiv:2211.03959 (2022).
> > > >
> > > > Schwarzer, M., Rajkumar, N., Noukhovitch, M., Anand, A., Charlin, L., Hjelm, R. D., Bachman,
> > > > P., & Courville, A. C. (2021b). Pretraining representations for data-efficient reinforcement
> > > > learning. Advances in Neural Information Processing Systems, 34, 12686–12699.

---

> > > > > ### Author Response · Authors · 2022-11-22
> > > > > **Follow-up to Response (part 1)**
> > > > >
> > > > > Dear Reviewer eTGE thank you for your reply.
> > > > >
> > > > >
> > > > > > Authors also suggest that this could be a first indication that pretraining helps in RL and Atari, but I would like to point out to the literature referred in Zhihui, et al. 2022 survey, and also works like Schwarzer et al. 2021 that give evidence on how pretraining helps in Atari
> > > > >
> > > > > Schwarzer et al. 2021 showed that pretraining helps in the low data regime (100k environment steps). Our fine-tuning experiments use 50M environment steps which represent 500x more data. It did not seem obvious in this setting that we would still benefit from pretraining on other variants. For example [1] found that using a pretrained representation as initialization led to faster learning but the benefits of the pretrained representation faded away after few hours of training (see Figure 5). Many other work on pretraining on Atari are limited to the Atari100k benchmark and it has been shown that performance after 100k steps may not be conclusive of performance after 200M frames / 50M steps (see Figure 8 from [2])
> > > > >
> > > > > Schwarzer et al. 2021 setting is also quite different from ours; they use the same game for pretraining (though without access to rewards) and fine-tuning, it is not too surprising that in the low data regime having access to (unlabeled) data from the same game is really helpful. In our case, using variants of the game means that our agents must generalize to benefit from pretraining. This has been shown to be a challenge for agents trained with reinforcement learning [3].
> > > > >
> > > > > The online pretraining section of Zhihui, et al. 2022 survey only covers unsupervised learning methods that interact with the environment when the reward function has not been specified yet. They mention that this is due to overfitting and that RL agents struggle to generalize,
> > > > > “Despite its primacy in providing excel performances for a specific task, the traditional RL paradigm faces two critical challenges when scaling it up to large-scale pretraining. **Firstly, it is notoriously easy for an RL agent to overfit (Zhang et al., 2018). As a result, a pretrained agent trained with sophisticated task rewards can hardly generalize to unseen task specifications**” (section 3). \
> > > > > They conclude with
> > > > > **“Despite its attractive effectiveness of learning without human supervision, online pretraining is still
> > > > > limited for large-scale applications.”** (section 4)
> > > > >
> > > > > In this work we show that policies trained with reinforcement learning on a few variants of a game are able to generalize on complex environments like Atari games.

---

> > > > > > ### Author Response · Authors · 2022-11-22
> > > > > > **Follow-up to Response (part 2)**
> > > > > >
> > > > > > > I was very surprised to read this statement given the fact that in the original IMPALA paper authors do experiment with a "deep IMPALA" -IMPALA with resnet- and "shallow IMPALA" with a CNN, and deep IMPALA shows much better performance in the Atari games.
> > > > > >
> > > > > > Our results do not contradict the original IMPALA paper, in our work we consider the “deep IMPALA” architecture as it is now standard and has been shown many times to perform better than the DQN architecture introduced by Mnih et al. (2015). This is made explicit in our paper, section 4.3:
> > > > > > “Our results so far hold for the network introduced with the IMPALA algorithm (Espeholt et al., 2018). Though it is an improvement over the DQN network architecture, the IMPALA network has only 15
> > > > > > layers and less than 2 million parameters.“
> > > > > >
> > > > > > Our results show that using networks deeper than the IMPALA networks with 15 layers can degrade performance. We agree the IMPALA network should perform much better than the original DQN architecture. The fact that the authors of the IMPALA paper designed a custom architecture with residual connections instead of one the networks introduced by [4] hints at the fact that they likely also observed a decrease in performance with deeper networks. This may also explain why the IMPALA architecture remains widely used to this day despite its small size compared with today’s standards.
> > > > > >
> > > > > > We show that RL agents do not scale with deeper / bigger architectures on Atari games when agents are initialized randomly. However these architectures can provide tremendous benefits after pretraining. Building on recent advances in vision and NLP, a lot of work has been done in recent years to scale RL with more data [5] and more compute [6-7]. However little work has been done on increasing the capacity of neural networks used in online reinforcement learning. **Our work shows that higher capacity networks can provide tremendous benefits and may be key to unlock large scale reinforcement learning**. We believe that this is an important result and will be of interest to the ICLR community:
> > > > > >
> > > > > > [1] Diversity is all you need: Learning skills without a reward function, Eysenbach et al. ICLR 2019 \
> > > > > > [2] Reincarnating Reinforcement Learning: Reusing Prior Computation to Accelerate Progress, Agarwal et al. NeurIPS 2022 \
> > > > > > [3] Generalization and regularization in DQN, Farebrother et al, arxiv preprint 2019. \
> > > > > > [4] Deep Residual Learning for Image Recognition, He et al. CVPR 2015 \
> > > > > > [5] MT-Opt: Continuous Multi-Task Robotic Reinforcement Learning at Scale, Kalashnikov et al. arxiv preprint 2021 \
> > > > > > [6] SEED RL: Scalable and Efficient Deep-RL with Accelerated Central Inference, Espeholt et al ICLR 2020 \
> > > > > > [7] Podracer architectures for scalable Reinforcement Learning, Hessel et al. arxiv preprint 2021 \
> > > > > > [8] A Dissection of Overfitting and Generalization in Continuous Reinforcement Learning, Zhang et al., arxiv preprint 2018

---

> > > > > > > ### Author Response · Authors · 2022-11-22
> > > > > > > **Follow-up to Response (part 3)**
> > > > > > >
> > > > > > > > Regarding the variants, I am afraid that without the promised update (right now it's is a conglomerate of letters and numbers without a mapping to understand them) we cannot assess what is the skill transfer evaluated here. This is important because in RL we cannot divide train and test tasks just randomly since the skill transfer that they may require could be impossible even for a human to do in zero-shot.
> > > > > > >
> > > > > > > We apologize for not providing the description of the variants earlier, we will add them to the paper in the next revision. The definition of the variants is available below with the design matrices for Space Invaders, Breakout and Asteroids available [here](https://imgur.com/a/1OgYaew).
> > > > > > >
> > > > > > > **Space Invaders**
> > > > > > >
> > > > > > > This is a shooting game where the player moves a laser cannon across the bottom of the screen and fires at aliens. The aliens begin on the upper half of the screen and move left and right as a group, they also shift downward each time they reach the edge of the screen. The player wins by eliminating all the aliens by shooting. The game ends with the aliens reaching the bottom or if the player loses its three lives. There are also 3 shields that offer temporary protection from the aliens’ cannon.
> > > > > > >
> > > > > > > The 32 game variants are defined by five discrete factors that modify the game:
> > > > > > > - The difficulty switch makes the player’s cannon wider and easier to hit. **(D)**
> > > > > > > - Shields move, it becomes harder to use them for protection. **(M)**
> > > > > > > - The aliens bombs zigzag making them harder to predict and avoid **(Z)**
> > > > > > > - The aliens laser drop faster **(F)**
> > > > > > > - The aliens are invisible, when one gets hit by the player the other become visible for short amount of time **(I)**
> > > > > > >
> > > > > > >
> > > > > > > **Breakout**
> > > > > > >
> > > > > > > Six rows of bricks are layered at the top half of the screen. Using a single ball the player must hit bricks using the paddle it controls at the bottom of the screen. The player must also keep as many lives as possible. If the player lets the ball fall below the bottom of the screen it loses a life. For each game a player gets five lives.
> > > > > > >
> > > > > > > The 24 variants are defined by the following parameters:
> > > > > > > - **(D)** The difficulty switch reduces the size of the paddle, making it harder to use to control the ball.
> > > > > > > - The rules the game:
> > > > > > >   * ( ) default rules
> > > > > > >   * **(T)** timed breakout, the player must break the wall as fast as possible, the number of lives used does not matter
> > > > > > >   * **(U)** Breakthru rules, Once the ball hits a brick, the ball continues to penetrate the wall, hitting more bricks and scoring more points
> > > > > > > - Extras:
> > > > > > >   * () default mode, the ball simply bounces off the paddle.
> > > > > > >   * **(S)** the player can steer the ball, it will follow the direction of the paddle.
> > > > > > >   * **(C)** when the ball makes contact with the paddle the player can press a button to keep the ball on the paddle. The player can wait to release the ball later at a more convenient location.
> > > > > > >   * **(I)** the wall becomes invisible and is only visible when a brick is hit.
> > > > > > >
> > > > > > >
> > > > > > > **Asteroids**
> > > > > > >
> > > > > > > In this game the player controls a ship in space that must destroy asteroids. The ship must avoid getting hit by asteroids, saucers will also appear regularly and try to shoot the ship
> > > > > > >
> > > > > > > The 63 variants are defined by the following factors:
> > > > > > > - The difficulty switch makes the game harder by introducing UFO and satellites. UFOs are hard to hit but are worth 1000 points when shot. Satellites are larger and easier to destroy. Both UFOs and Satellites will be firing back.
> > > > > > > - **(F/S)** asteroids will be moving fast or slowly.
> > > > > > > - Extra lives
> > > > > > >   * **5**: one additional life every 5000 points
> > > > > > >   * **10**: one additional life 10,000 points
> > > > > > >   * **20**: one additional life very 20,000 points
> > > > > > >   * **N**: no additional lives
> > > > > > > - Features
> > > > > > >   * **(H)** hyperspace: the player can make its spaceship disappear and reappear at another place in the screen. This can be useful to get away from incoming enemies but the ship can reappear on the path of another asteroid.
> > > > > > >  * **(SH)** protective shields can be set up by the player to protect its ship however keeping shields up for more than 2 seconds  will blow up the ship
> > > > > > >   * **(FL)** Flip in, it becomes possible to flip the spaceship 180 degrees to quickly aim in the opposite direction
> > > > > > >  * **(W)** no optional features
> > > > > > >
> > > > > > > Mode 128 (the 33rd mode) is for young children, the game features slow asteroids, hyperspace, and an extra ship each 5000 points.

---

> > > > > > > > ### Author Response · Authors · 2022-11-22
> > > > > > > > **Follow up to Response (part 4)**
> > > > > > > >
> > > > > > > > **Bank Heist**
> > > > > > > >
> > > > > > > > This is a maze game similar to Pac-Man. The objective is to rob as many banks
> > > > > > > > as possible while avoiding police cars that appear every time a bank is robbed. The player controls a car called a getaway car with limited fuel. When a bank is robbed a police car and a new bank both appear. Police cars can be destroyed by dropping dynamite from the getaway. The player starts with four lives and can lose a life by running out of fuel, getting hit by dynamite or getting hit by a police car. The car is refuelled every time the player changes cites.
> > > > > > > >
> > > > > > > > There are 32 variants defined by the following factors
> > > > > > > > - The difficulty is set through two switches
> > > > > > > >   * Left difficulty switch A:  Police cars are smart and follow the player’s car.
> > > > > > > >   * Left difficulty switch B:  Police cars are not smart and follow a random pattern after appearing.
> > > > > > > >   * Right difficulty switch A:  Banks appear in a random order.
> > > > > > > >   * Right difficulty switch B:  Banks appear at preset locations
> > > > > > > > - These two switches lead to 4 difficulty levels
> > > > > > > >   * **(0)**: Left B, Right B
> > > > > > > >   * **(1)**: Left A, Right B
> > > > > > > >   * **(2)**: Left B, Right A
> > > > > > > >   * **(3)**: Left A, Right A
> > > > > > > > - There are 8 modes, each mode corresponds to a level with four cities. The game ends after leaving the fourth town. In higher levels the pursuit is faster, the car burns fuel faster and there is less time to rob banks.

---

### Author Response · Authors · 2022-11-15
**Rebuttal: summary of changes**

We would like to thank the reviewers, $R1$ (eTGE),  $R2$ (t4n9),  $R3$ (JzdD) and  $R4$ (9aV5 ) for taking the time to read our manuscript and provide valuable feedback. We have uploaded the new revision:
- We found an issue in our evaluation protocol that affected our results but our main findings are largely unaffected. Because of that the reported results for zero-shot evaluation (Figure 2) were lower than they should be. This issue was also hidden by the fact that we were reporting training curves (where the policy can significantly change over the course of an episode using distributed training) instead of evaluation curves.
    - We now report evaluation curves using a separate evaluation process, the evaluation policy is kept fixed for the duration of an episode.
    - To evaluate the performance of zero shot transfer and transfer after fine-tuning for 200 million frames each policy is evaluated for 50 episodes per seed.
    - We also rerun all our experiments, using 10 seeds instead of five previously for fine-tuning experiments. Pretraining is still done using only one seed due to the cost of pretraining for 15 billion frames which is 75x greater than fine-tuning..
- We took into account feedback from the reviewers and **made the following changes to the paper**
    - We added experiments using only one or two variants for pretraining in **Figure 6**.  [$R3$]
    - Changed the y-axis for Figure 6 from log scale to linear scale. [$R3$]
    - *Training curves during multi task pretraining*: Added the training curves for pretraining experiments with 29 modes on Space Invaders and 32 modes on Asteroids in **Figure 10 and 11 in the appendix**.  [$R2$]
    - *Instability of residual networks for online RL on Atari*: We added experiments showing that instabilities observed when training residual networks without pretraining are not limited to Asteroids and Space Invaders **in Figure 8 in the appendix**. [$R2$]
    - *IMPALA-CNN scaling in width*: Added a comparison with Cobbe et al. scaling of the IMPALA-CNN architecture in width by increasing the number of channels in convolutional layers, **in Appendix A.1 / in Figure 7**. [$R3$].

---

### Decision · Program_Chairs · 2023-01-20

**Decision:**

Accept: poster

**Justification For Why Not Higher Score:**

The experiments are limited in Atari.

**Justification For Why Not Lower Score:**

The experimental results on pretraining are valuable to the community

**Metareview: Summary, Strengths And Weaknesses:**


This paper studies multi-task pretraining on a set of atari game variants followed by fine-tuning on an unseen variant. The paper shows that while multi-task pretraining often doesn't help zero-shot performance, it improves the fine-tuning performance and sample efficiency significantly, and these improvements scale as pretraining dataset size and the number of variants increases. Furthermore, it was shown that in this multi-task pretraining regime, larger models than previously used can provide strong benefits to both pretraining and fine-tuning performance.
The reviewers raised concerns about the claims and experiment setups. The AC thanks the authors and reviewers for extensive and engaging discussions.
The AC believes the paper presents rigorous studies on several phenomena about pertaining, which are valuable to the community.

**Note From Pc:**

if the above contains the word "oral" or "spotlight" please see: "oral" presentation means -> notable-top-5% and "spotlight" means -> notable-top-25%. As stated in our emails, we are disassociating presentation type from AC recommendations